# Medical image encryption algorithm based on a new five-dimensional three-leaf chaotic system and genetic operation

**Zhongyue Liang**[1,2]**, Qiuxia Qin**[1,2]**, Changjun Zhou**[3]***, Ning Wang**[4]**, Yi Xu**[4]***,
**Wenshu Zhou**[4]

**1** Department of Computer Science and Engineering, Dalian Minzu University, Dalian, China, **2** Institute of Machine Intelligence and Biological Computing, Dalian Minzu University, Dalian, China, **3** College of Mathematics and Computer Science, Zhejiang Normal University, Jinhua, China, **4** Department of Science & Department of Pre-university, Dalian Minzu University, Dalian, China

\* zhou-chang231@163.com (CZ); xuyi@dlnu.edu.cn (YX)

**Data Availability Statement:** All relevant data are in the manuscript, and the picture information is in the Figure.

## Abstract

Current image encryption methods have many shortcomings for the medical image encryption with high resolution, strong correlation and large storage space, and it is difficult to obtain reliable clinically applicable medical images. Therefore, this paper proposes a medical image encryption algorithm based on a new five-dimensional three-leaf chaotic system and genetic operation. And the dynamic analysis of the phase diagram and bifurcation diagram of the five-dimensional three-leaf chaotic system selected in this paper is carried out, and NIST is used to test the randomness of its chaotic sequence. This algorithm follows the diffusion-scrambling framework, especially using the principle of DNA recombination combined with the five-dimensional three-leaf chaotic system to generate a chaotic matrix that participates in the operation. The bit-level DNA mutation operation is introduced in the diffusion, and the scrambling and diffusion effects have been further improved. Algorithm security and randomness have been enhanced. This paper evaluates the efficiency of this algorithm for medical image encryption in terms of security analysis and time performance. Security analysis is carried out from key space, information entropy, histogram, similarity between decrypted image and original image, PSNR, correlation, sensitivity, noise attack, cropping attack and so on. Perform time efficiency analysis from the perspective of time performance. The comparison between this algorithm and the experimental results obtained by some of the latest medical image encryption algorithms shows that this algorithm is superior to the existing medical image encryption algorithms to a certain extent in terms of security and time efficiency.

## 1 Introduction

In recent years, with the rapid development of the Internet, the rapid expansion of information, a large amount of information is transmitted in public channels, and the issue of information security is followed by [1–3]. There are many kinds of information data, such as text,

**Funding:** This research was funded by the National Youth Science Foundation of China No. 62002046 and No. 61802040. Among them, the funds provided by the National Youth Science Fund No. 62002046 are used for investigation and research, paper writing and publishing funds for this research. The funds provided by the National Youth Science Fund No. 61820040 are used for methodology and paper writing. Moreover, this research was not supported by other funds. In addition, explain the role of funders in the research. The funder Changjun Zhou played a role in study design and preparation of the manuscript. The funder Yi Xu played a role in data collection and analysis, and preparation of the manuscript. The funder Wenshu Zhou played a role in data collection and analysis.

**Competing interests:** The authors have declared that no competing interests exist.

language, pictures, videos, etc., among which pictures are the most widely used in information transmission [4]. The protection of pictures has attracted the attention of many researchers. Among various protection measures of pictures, image encryption occupies an important position [5, 6]. Therefore, on the basis of many image encryption algorithms, it is necessary to study more real-time, efficient and safe encryption algorithms.

Medical image is a kind of image information, which reflects the patient's physical condition and belongs to some private information of the patient, such as MRI, CT and other medical images [7]. Compared with ordinary images, these medical images have higher redundancy, larger data volume and stronger correlation [8]. For this special image encryption, encryption algorithm security and real-time requirements will be more stringent.

Chaos is a deterministic nonlinear system that can generate seemingly random but inherently ordered chaotic sequences [9]. The characteristics of chaos, such as initial value sensitivity, parameter sensitivity, and internal randomness [10], have a natural relationship with cryptography. The introduction of a chaotic system into the encryption algorithm improves the randomness of the encryption algorithm. Chaotic systems are divided into low-dimensional chaotic systems [11] and high-dimensional chaotic systems [12, 13]. In the early days, researchers only used low-dimensional chaotic systems, such as Logistic chaos [14], and put them into the encryption algorithm. Although it is better than the encryption algorithm that does not use chaos, it is simple and easy to implement, but the key space is small and easy to be cracked by brute force; Therefore, researchers improved the system based on the characteristics of the Logistic chaotic system and introduced it into the encryption algorithm to achieve better results [8, 15]; In addition, Bao et al. [16] proposed the use of discrete mHR model and ISI coding to generate chaotic sequences for image encryption. The novel idea opens up new perspectives in the field of image encryption. However, the use of chaotic systems for image encryption is still the mainstream direction of researchers. Therefore, this paper still uses a high-dimensional chaotic system that can increase the key space, and introduces a new five-dimensional three-leaf chaotic system that can generate three-leaf chaotic attractors in multiple directions [17], which significantly improves the randomness of chaotic sequences and effectively resists violent attacks.

DNA contains a large amount of genetic information, which is the necessary genetic material for various living organisms. DNA molecule is composed of 4 nitrogen-containing bases, and a DNA molecular sequence can carry huge information [18]. Due to the shortcomings of traditional electronic computer such as small storage capacity and serial connection, the combination of DNA and computer technology is a hot spot in cryptography research [5]. In recent years, researchers use computers instead of traditional biochemical experiments to achieve DNA manipulation, which is called pseudo-DNA manipulation [19, 20]. The DNA molecule has the characteristics of large storage capacity and strong parallelism [21, 22], which can be well applied to the image encryption field with huge amount of information and high real-time requirements, including DNA encoding [23], DNA computing [24] and DNA decoding [25] etc. In the early days, DNA coding rules were fixed [26], DNA calculation rules were fixed and there were few types [27–30], which could be easily cracked by brute force; Therefore, this article was changed to bit-level dynamic DNA coding [31, 32]. Common DNA operations such as addition and subtraction [27–30] are expanded to 7 types of DNA operations. The randomness is improved, and the diffusion effect is significantly enhanced.

DNA mutation [33] refers to a change in DNA base or a change in base sequence, and base substitution mutation is a type of DNA mutation. As we all know, DNA bases [5] mainly include two kinds of purines and two kinds of pyrimidines. Base substitution mutation [34] is divided into transition and transversion. Transition is the exchange of two different purines or

two different pyrimidines; Transversion is the exchange of one purine and one pyrimidine. The occurrence of DNA mutations is affected by a specific combination of genes, and this mutation can be said to be a random mutation. This kind of random base substitution mutation occurs during the complementary pairing of two kinds of DNA. Incorporating this DNA mutation operation [35, 36] into the encryption algorithm can change the pixel value more fundamentally, and better meet the requirements of high randomness and large change rate of image information encryption. Therefore, the DNA mutation operation not only increases the biological significance of image encryption, but also effectively improves the robustness and security of the encryption algorithm.

DNA recombination technology [37–39] refers to the use of artificial means to recombine DNA fragments with a certain special meaning from different sources to achieve the purpose of changing biological characteristics. In this paper, DNA recombination technology is introduced into DNA encryption, and arbitrary fragments are changed by artificial means. The DNA fragments are exchanged immediately, and new gene combinations can be generated. In this article, the DNA recombination operation is to further complete the diffusion and scrambling of the pixels, which greatly increases the difficulty of the algorithm to decipher. Therefore, the DNA recombination operation not only increases the complexity of the DNA sequence, but also improves the randomness and security of the encryption system.

In summary, this paper proposes a medical image encryption algorithm based on a new five-dimensional three-leaf chaotic system and genetic operation. The remaining sections of the paper are arranged as follows: Section 2 mainly introduces the basic principles used in the algorithm; Section 3 mainly introduces the specific methods and steps of the algorithm; Section 4 mainly uses simulation software for security testing; Finally, a brief summary of this algorithm.

## 2 Basic principles

### 2.1 Logistic chaotic map

Logistic chaotic map [14, 15] is a typical one-dimensional chaotic system. Although the definition is simple, it has complex random dynamic behavior and can generate random chaotic sequences. The definition is as shown in Eq (1).

$$x_{n+1} = \mu x_n (1 - x_n) \tag{1}$$

Where $\mu$ is the control parameter. When $0 < \mu \leq 4$, $0 < x < 1$, the system is in a state of chaos. The bifurcation diagram of the Logistic chaotic map is shown in Fig 1.

### 2.2 Five-dimensional three-leaf chaotic system

**2.2.1 The definition of chaotic system and its dynamic characteristics analysis.** In 2020, Wang et al. [17] introduced two controllers $w$ and $v$ based on the three-dimensional chaotic system proposed by Sprott [40], feedback $w$ to the original controller $y$, and feedback $y$, $z$ to the new controller $w$, feedback $v$ to $w$, feedback $x$, $y$ to the new controller $v$. The interaction of the five controllers makes the parameter control of the system more complicated, and can generate three-leaf chaotic attractors in multiple directions. This new type of five-dimensional

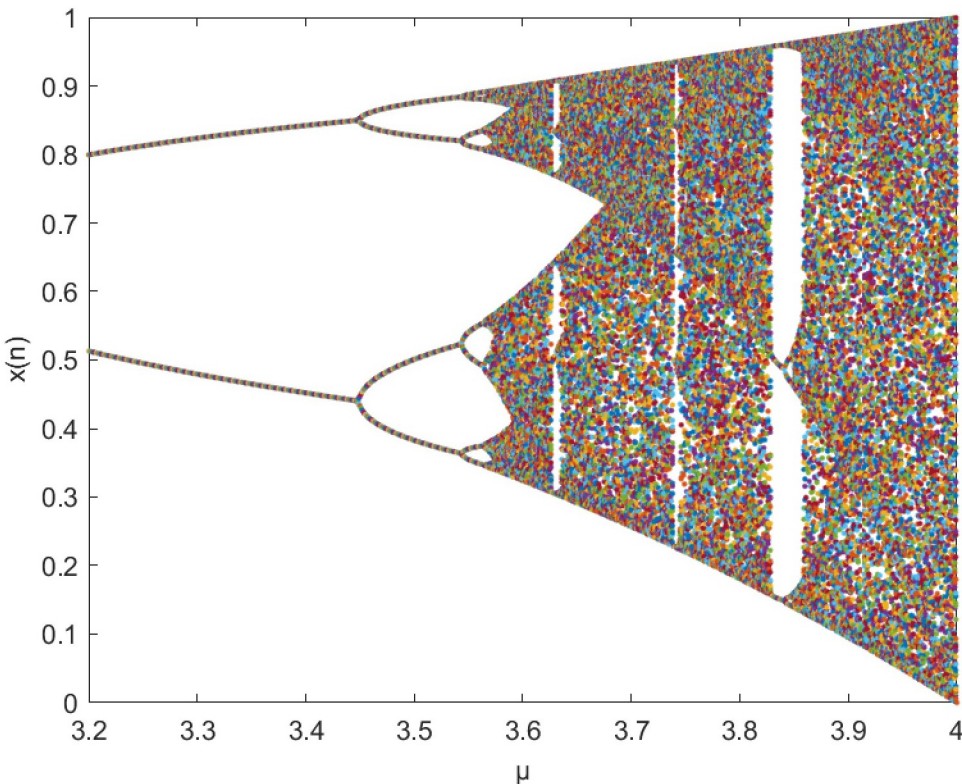

**Fig 1. Logistic chaotic bifurcation diagram.**

three-leaf chaotic system is defined as shown in Eq (2).

$$\begin{cases} \dot{x} = ax + byz^2 \\ \dot{y} = x + z^2 - \omega z \\ \dot{z} = 1 + x \\ \dot{\omega} = yz + c\omega + dv \\ \dot{v} = xy + ev \end{cases} \tag{2}$$

Where $a$, $b$, $c$, $d$, $e$ are the control parameters, when $a = -1$, $b = -3$, $c = -3$, $d = 1.8$, $e = -5$, the system is in a chaotic state.

Under normal circumstances, the higher the dimension of the chaotic system, the better the security of the entire encryption algorithm can be guaranteed. Moreover, compared with low-dimensional chaotic systems, high-dimensional chaotic systems can provide chaotic sequences with stronger randomness, obtain a larger key space, and improve the security of the entire encryption algorithm. But there are also some shortcomings. For example, the higher the dimensionality of the chaotic system, the more complicated the operation will be. However, the five-dimensional three-leaf chaotic system is simpler and easier to implement than most high-dimensional chaotic systems. In order to prove the dynamic characteristics of the chaotic system, this paper shows the three-dimensional phase diagram of the chaotic system, as shown in Fig 2.

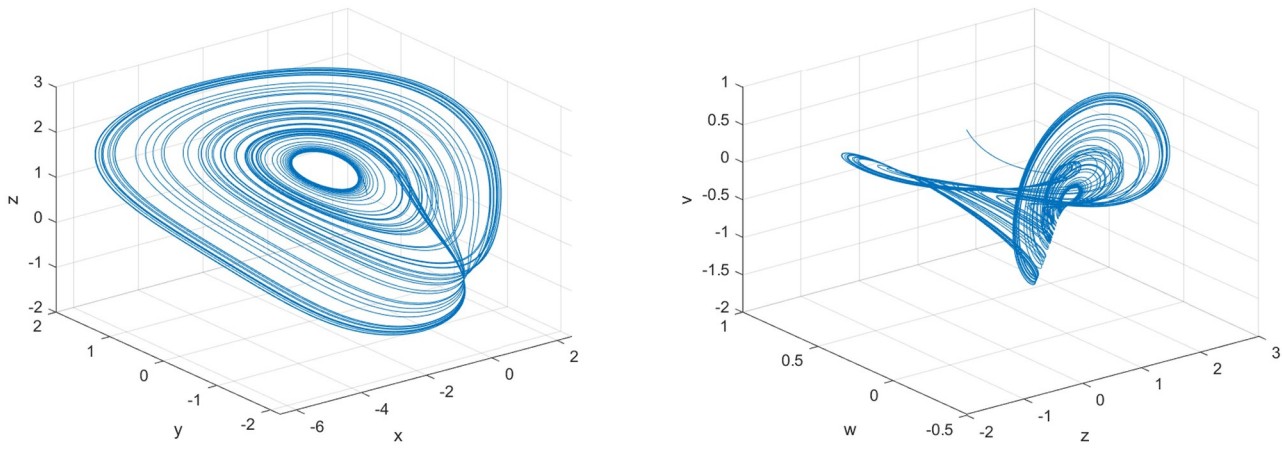

**Fig 2. Phase diagram of the new five-dimensional three-leaf chaotic system.** (a) *x-y-z*, (b) *z-w-v*.

In addition, Wang et al. [17] carried out a series of dynamic analyses such as balance points and bifurcation diagrams while proposing the chaotic system. The experimental results proved that the chaotic system has strong stability. In the corresponding interval, the bifurcation diagrams of parameter a and parameter b show chaotic state. In addition, Wang et al. also applied the system design to image encryption and achieved good encryption results. Therefore, compared with most other chaotic systems, the structure of this five-dimensional three-leaf chaotic system is simpler, and the dynamic characteristics are more obvious, and the effect is significant in practical applications.

**2.2.2 Chaotic sequence randomness test.**   The above section analyzes the dynamic characteristics of the five-dimensional three-leaf chaotic system. In practical applications, it is the chaotic sequence generated by the chaotic system that is directly applied to the encryption algorithm. Although the dynamic characteristics of the chaotic system is very important when selecting the chaotic system. However, it is more important to select a chaotic system that can generate a chaotic sequence with strong randomness. This section uses the randomness test tool (SP800-22) to test the randomness of the chaotic sequence from 15 aspects. The test results are shown in Table 1.

After the NIST test is completed, the probability value obtained when counting the results is also called the P value. When the P value of a test is greater than or equal to 0.01, it indicates that the randomness test meets the standard. It can be seen from the table that the chaotic sequence generated by the chaotic system has passed all random tests. And this system is better than the NIST test results of chaotic sequences generated by general high-dimensional chaotic systems. For example, the P value of each item is slightly higher than the P value of 15 tests in [41], which means it has better randomness. In addition, Wang et al. [17] also tested the time series diagrams in each direction of *x*, *y*, *z*, *w*, and *v*, and they all showed a chaotic state, which also shows that the chaotic sequence generated by this system is more random. The security of the image encryption algorithm is more guaranteed.

## 2.3 DNA coding and operations

DNA contains 4 kinds of bases [5], namely A (adenine), T (thymine), C (cytosine), and G (guanine). Traditional computer processing of data is usually expressed in binary form of 0 and 1.

**Table 1. NIST test result display table of chaotic sequence.**

| Test index | P-value | Ref. [41] P-value | Result |
|---|---|---|---|
| Frequency | 0.740505 | 0.727840 | pass |
| Block-frequency | 0.620972 | 0.492885 | pass |
| Cumulative Sums | 0.471003 | 0.467379 | pass |
| Runs | 0.610857 | 0.539295 | pass |
| Longest Run of Ones | 0.784304 | 0.503831 | pass |
| Rank | 0.738246 | 0.719200 | pass |
| FFT | 0.711127 | 0.358795 | pass |
| Non Periodic Template | 0.440431 | 0.234494 | pass |
| Overlapping Template | 0.166231 | 0.157774 | pass |
| Universal | 0.839712 | 0.607603 | pass |
| Approximate Entropy | 0.893543 | 0.838764 | pass |
| Random Excursions | 0.840598 | 0.799237 | pass |
| Random Excursions Variant | 0.952241 | 0.949623 | pass |
| Linear Complexity | 0.672006 | 0.640112 | pass |
| Serial | 0.643920 | 0.276930 | pass |

**Table 2. DNA coding rules.**

| Binary | R1 | R2 | R3 | R4 | R5 | R6 | R7 | R8 |
|---|---|---|---|---|---|---|---|---|
| 00 | A | A | C | C | G | G | T | T |
| 01 | C | G | A | T | A | T | C | G |
| 10 | G | C | T | A | T | A | G | C |
| 11 | T | T | G | G | C | C | A | A |

If a 2-bit binary is used to represent a base, there are 24 different representation methods in total [42]. Based on the principle of base complementary pairing [43–45], A and T are complementary, and C and G are complementary, so there are 8 kinds of expressions that conform to the principle, called 8 kinds of DNA coding rules, as shown in Table 2.

In addition to DNA coding rules, based on binary calculations, DNA addition [27, 28], DNA subtraction [29], DNA XOR [30], DNA XNOR, DNA multiplication, DNA shift left and DNA shift right operation rules are given. As shown in Tables 3–9.

## 2.4 DNA mutation

DNA mutation [33, 34] refers to changes in the position and sequence of the nitrogenous bases that make up the DNA molecule. There are 4 types of DNA mutations [35], which are base substitution mutations, frame shift mutations, in-frame mutations, and chromosomal

**Table 3. DNA addition.**

| ADD | A | C | G | T |
|---|---|---|---|---|
| A | C | A | G | T |
| C | A | C | G | T |
| G | T | G | C | A |
| T | G | T | A | C |

**Table 4. DNA subtraction.**

| SUB | A | C | G | T |
|---|---|---|---|---|
| A | C | G | T | A |
| C | A | C | G | T |
| G | T | A | C | G |
| T | G | T | A | C |

**Table 5. DNA XOR.**

| XOR | A | C | G | T |
|---|---|---|---|---|
| A | A | C | G | T |
| C | C | A | T | G |
| G | G | T | A | C |
| T | T | G | C | A |

**Table 6. DNA XNOR.**

| XNOR | A | C | G | T |
|---|---|---|---|---|
| A | T | G | C | A |
| C | G | T | A | C |
| G | C | A | T | G |
| T | A | C | G | T |

**Table 7. DNA multiplication.**

| MUL | A | C | G | T |
|---|---|---|---|---|
| A | T | G | C | A |
| C | G | T | A | C |
| G | C | A | T | G |
| T | A | C | G | T |

**Table 8. DNA shift left.**

| LSHIFT | A | C | G | T |
|---|---|---|---|---|
| A | A | T | G | C |
| C | C | A | T | G |
| G | G | C | A | T |
| T | T | G | C | A |

**Table 9. DNA shift right.**

| RSHIFT | A | C | G | T |
|---|---|---|---|---|
| A | A | C | G | T |
| C | C | G | T | A |
| G | G | C | A | T |
| T | T | A | C | G |

mutations. Among them, base substitution mutation can be specifically divided into transition and transversion [36]. Transition is defined as the replacement of one purine by another purine, or the replacement of one pyrimidine by another pyrimidine, and transversion is defined as the replacement of one purine by another pyrimidine or the replacement of one pyrimidine by another purine.

Based on the principle of base complementary pairing, a single-base direct complementary pairing rule is defined [44, 45], which is called complementary rule 1 hereinafter, and its operation is called complementary operation 1, as shown in Eq (3).

$$
\begin{cases}
T = complement(A) \\
A = complement(T) \\
C = complement(G) \\
G = complement(C)
\end{cases}
\tag{3}
$$

Where the *complement*(.) is a complementary function, for example, the complement of base A is T, and the complement of base C is G.

In addition, based on the double helix structure of DNA, the following principles of complementary pairing based on single-base and double-base are defined [43–45], hereinafter referred to as complementary rule 2, and its operation is referred to as complementary operation 2, as shown in Eq (4).

$$
\begin{cases}
x_i \neq D(x_i) \neq D(D(x_i)) \neq D(D(D(x_i))) \\
x_i = D(D(D(D(x_i))))
\end{cases}
\tag{4}
$$

Among them, $x_i$ and $D(x_i)$ are complementary. The principle of complementarity can realize the principle of base substitution mutation, which is used to realize the mutation of a single base. The specific rules are shown in Table 10.

When performing DNA mutations, first dynamically select the DNA mutation rules and the number of DNA mutations to increase the diversity and flexibility of DNA mutations, and then use complementation rule 1 and complementation rule 2 to achieve single-base mutations. To a certain extent, the random replacement of a single base is realized, which in turn changes the value of the corresponding pixel value, and also realizes the diffusion of the pixel level. Therefore, applying DNA mutation to this encryption algorithm has a certain meaning and effect.

**Table 10. DNA mutation.**

| Rules | DNA mutation operation | | | |
|-------|------|------|------|------|
| RR1 | (AT) | (TC) | (CG) | (GA) |
| RR2 | (AT) | (TG) | (GC) | (CA) |
| RR3 | (AC) | (CG) | (GT) | (TA) |
| RR4 | (AC) | (CT) | (TG) | (GA) |
| RR5 | (AG) | (GC) | (CT) | (TA) |
| RR6 | (AG) | (GT) | (TC) | (CA) |

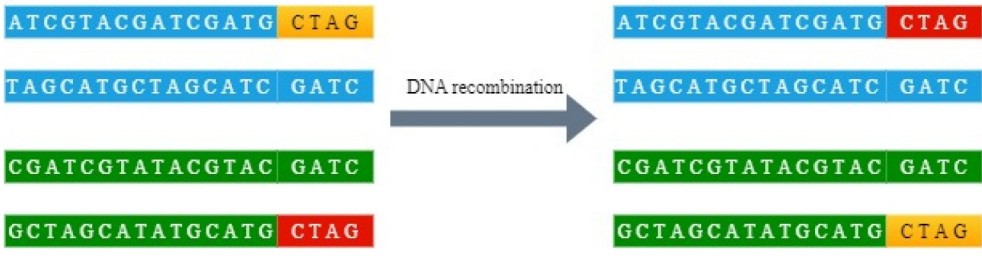

**Fig 3. Schematic diagram of DNA recombination.**

## 2.5 DNA recombination

DNA recombination [37–39] refers to the recombination of DNA fragments from different sources to disrupt the original DNA sequence. For example, the two DNA double strands intercept a 4-base DNA fragment at the end, and recombine the two DNA fragments with the original DNA strand to obtain two DNA strands that are different from the original. As shown in Fig 3.

During DNA recombination, the position of base exchange between two DNA sequences is artificially controlled, and DNA recombination operations are performed on the two DNA sequences. Part of the base sequence on the DNA sequence is replaced, which improves the scrambling degree and diffusion effect of the image, and the degree of scrambling is still bit level, so this article applies the principle of DNA recombination to the image encryption algorithm.

## 3 Encryption and decryption algorithm

### 3.1 Encryption algorithm

This algorithm mainly includes the following steps: Firstly, the known key stream and the original image respectively use hash functions to generate digest information, and then undergo a series of processing to generate the final key stream. The final key stream passes through the five-dimensional three-leaf chaotic system and combines the principle of DNA recombination to generate a chaotic matrix; Secondly, the original image is subjected to bit-level dynamic DNA coding; Then, index scrambling is performed to change the pixel position; Then, the DNA coding matrix and chaos matrix are used to perform dynamic DNA operations, and the principle of DNA mutation is used to achieve pixel diffusion; Finally, DNA dynamic decoding is performed to get the decrypted image. The specific implementation process and steps are given in Section 3.2-3.7 and Fig 4.

### 3.2 Key generation

This article inputs a 128-bit key stream *KeyHex* for encryption, the original image to be encrypted image is *I*, *M* and *N* are the length and width of the image respectively.

**3.2.1 Generation of key stream.** This algorithm mainly uses MD5 and SHA-256 two algorithms to generate the final key stream.

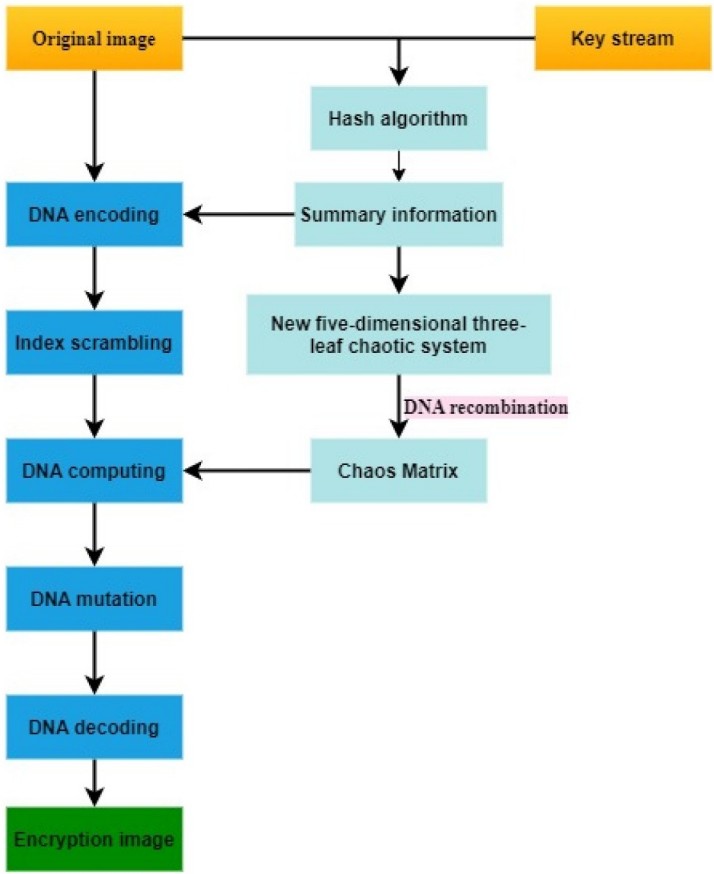

**Fig 4. Encryption flowchart.**

Step 1: This algorithm inputs the original image and the *KeyHex* key stream, and calculates the sum of row pixel values and the sum of column pixel values, as shown in Eq (5).

$$\begin{cases} SumRow = sum(I, 2) \\ SumCoL = sum(I, 1) \end{cases} \tag{5}$$

Among them, *I* is the original image, and *sum*() calculates the sum of pixel values.

Step 2: The MD5 algorithm in the hash algorithm is called three times to generate data stream *D* as an intermediate data stream. The generation process is as shown in Eq (6).

$$D = [hash(SumRow,'MD5'), hash(SumCoL,'MD5'), hash(KeyHex,'MD5')] \tag{6}$$

Among them, *hash*() is a hash function.

Step 3: Use the SHA-256 algorithm in the hash algorithm to generate a 128-bit key stream *H*. The generation process is as shown in Eq (7).

$$H = hash(D,'SHA - 256') \tag{7}$$

Step 4: Convert every 4 bits in the 128-bit binary key stream into 1 decimal, and the key stream is converted into a 32-bit decimal key stream *HexDecimal*. The 128-bit *KeyHex* is also converted into a 32-bit decimal key stream *HashDecimal*.

Perform a bitwise XOR operation on *HexDecimal* and *HashDecimal* to generate the final 32-bit decimal key stream *Key*. The generation process is shown in Eq (8).

$$Key = bitxor(HexDecimal, HashDecimal) \tag{8}$$

Among them, $bitxor()$ is a bitwise XOR operation.

**3.2.2 Generation of key feature values.** Each bit of the generated 32-bit decimal key stream *Key* is XOR to each other to generate a decimal key feature value *KeyFeature*. The generation process is as shown in Eq (9).

$$KeyFeature = bitxor(KeyFeature(i), KeyFeature(i+1)) \tag{9}$$

Among them, $1 \le i \le 32$, $bitxor()$ is a bitwise XOR operation.

## 3.3 Generation of chaotic matrix

This algorithm uses a five-dimensional three-leaf chaotic system, a one-dimensional Logistic chaotic system and the principle of DNA recombination to generate a chaotic matrix.

**3.3.1 Generation of five-dimensional three-leaf chaotic matrix.**

Step 1: Use the first 30-bit key stream of *Key* and the key feature value *KeyFeature* to generate the initial values of the five-dimensional three-leaf chaotic system $x(1)$, $y(1)$, $z(1)$, $w(1)$, $v(1)$. The production process is as shown in Eq (10).

$$\begin{cases} x(1) = (KeyFeature \oplus K(1) \oplus K(2) \oplus K(3) \oplus K(4) \oplus K(5) \oplus K(6))/256 \\ y(1) = (KeyFeature \oplus K(7) \oplus K(8) \oplus K(9) \oplus K(10) \oplus K(11) \oplus K(12))/256 \\ z(1) = (KeyFeature \oplus K(13) \oplus K(14) \oplus K(15) \oplus K(16) \oplus K(17) \oplus K(18))/256 \\ w(1) = (KeyFeature \oplus K(19) \oplus K(20) \oplus K(21) \oplus K(22) \oplus K(23) \oplus K(24))/256 \\ v(1) = (KeyFeature \oplus K(25) \oplus K(26) \oplus K(27) \oplus K(28) \oplus K(29) \oplus K(30))/256 \end{cases} \tag{10}$$

Among them, $K(.)$ is used to replace $Key(.)$

Step 2: In order to improve the randomness, it is necessary to discard the chaotic sequence generated in the early stage of the chaotic system. Use the 31-bit, 32-bit and *KeyFeature* of *Key* to generate the number of iterations that need to be discarded *discard*. The generation process are shown in Eq (11).

$$discard = Key(31) + Key(32) + KeyFeature \tag{11}$$

Step 3: This algorithm uses a five-dimensional chaotic system. One iteration of the system will produce 5 values, so the total number of iterations *SizeSignal* is calculated as shown in Eq (12).

$$SizeSignal = 4ceil(MN/5) + discard \tag{12}$$

Among them, *ceil*() is a function for rounding up, and *M* and *N* are the length and width of the image.

Step 4: The chaotic system discards the result of the previous *discard* iteration, and continues to perform *SizeSignal − discard* iterations to generate a 4*MN* size *matrix*1 chaotic matrix.

### 3.3.2 Logistic chaotic matrix generation.

Step 1: Use the key stream *Key* and the key feature value *KeyFeature* to generate the control parameter *μ* of the Logistic chaotic system. The generation process is shown in Eqs (13) and (14).

$$xx = (KeyFeature \oplus K(2) \oplus K(3) \oplus K(4) \oplus K(5) \oplus K(6) \oplus K(7) \oplus K(8) \oplus K(9))/256 \quad (13)$$

$$\mu = 3.89 + 0.01 * xx \quad (14)$$

Among them, *K*(.) is used to replace *Key*(.)

Step 2: Use the key stream *Key* and the key feature value *KeyFeature* to generate the initial value *a*(1) of Logistic chaos. The generation process are shown in Eq (15).

$$a(1) = (KeyFeature \oplus K(10) \oplus K(11) \oplus K(12)\oplus$$
$$K(13) \oplus K(14) \oplus K(15) \oplus K(16) \oplus K(17))/256 \quad (15)$$

Among them, *K*(.) is used to replace *Key*(.)

Step 3: Using the control parameters and initial values generated above, the Logistic chaotic system iterates 4*MN* times to generate a *matrix*2 chaotic matrix with a size of 4*MN*.

### 3.3.3 The principle of DNA recombination generates the final chaotic sequence. Use the principle of DNA recombination to generate the final chaotic matrix *KeyMatrix* with a size of 4*MN*. The generation process is shown in Eqs (16) and (17).

$$matrix1(2MN + 1 : 4MN) = matrix2(2MN + 1 : 4MN) \quad (16)$$

$$KeyMatrix = matrix1 \quad (17)$$

## 3.4 Bit-level dynamic DNA coding

### 3.4.1 Pretreatment before DNA coding. The size of the original image *I* is *MN*, which corresponds to a matrix of *MN* size. Each elements of matrix is a decimal pixel value. One decimal pixel value is converted into 8-bit binary, and every two digits are converted into a base according to a certain DNA coding rule. Basically, at this time, the size of the matrix becomes a 4*MN*-sized DNA matrix, so it is necessary to convert the *MN*-sized matrix to a 4*MN* size in advance to store the DNA-encoded matrix.

### 3.4.2 DNA coding.

Step 1: Generate DNA coding rules, and use one DNA coding rule for every 2-bit binary, so the length of DNA coding rule should be 4*MN* size. Using the Logistic chaotic sequence,

the key stream *Key* and the key feature value *KeyFeature* generate a chaotic sequence *LogisticSeq* with a length of 4*MN*, and the DNA encoding rule *R* is generated as shown in Eq (18).

$$R = floor(8 * LogisticSeq) + 1 \qquad (18)$$

Among them, *floor*() is a round-down function.

Step 2: Circulate every 2-bit binary in each pixel value, and generate the corresponding bases according to the corresponding DNA coding rules in *R* as shown in the Fig 5. When all the pixel values are encoded, a DNA encoding matrix *DecodedMatrix* with a size of 4*MN* is generated.

## 3.5 Index scrambling

Index scrambling is an operation that uses a chaotic sequence to scramble the position of each base in the matrix.

Step 1: According to 3.3, the size of the DNA matrix is 4*MN*. Use Logistic chaotic sequence, key stream *Key* and key feature value *KeyFeature* to generate chaotic sequence *ChaoticSignal* with a length of 4*MN*.

Step 2: Use *ChaoticSignal* to generate the index sequence *Pos*, and use the index sequence to scramble the DNA coding matrix *DecodedMatrix* to generate the scrambled DNA matrix *PerMatrix*. The production process is as shown in Eqs (19) and (20).

$$[ \ , Pos] = sort(ChaoticSignal) \qquad (19)$$

$$PerMatrix = DecodedMatrix(Pos) \qquad (20)$$

Among them, the *sort*() function is to sort the elements.

## 3.6 Dynamic DNA operations

DNA operation is to operate the DNA matrix *PerMatrix* that has been scrambled and the chaotic matrix *KeyMatrix* generated in Section 3.2, use the chaotic sequence to generate DNA operation rules, determine which DNA operation method the two perform, and then perform the DNA operation to generate a new matrix process.

Step 1: Using Logistic chaotic sequence again, the key stream *Key* and the key feature value *KeyFeature* generate the chaotic sequence *ChaoticSignal*1 with a length of 4*MN*. Because DNA has 7 operation methods, the following methods are used to generate the DNA operation rule *Operation*. The generation process shown in Eq (21).

$$Operation = floor(7 * ChaoticSignal1) + 1 \qquad (21)$$

Among them, *floor*() is a round-down function.

Step 2: Each elements of matrix of *PerMatrix* and *KeyMatrix* selects the DNA operation method according to the operation rules of *Operation*, and performs DNA operation.

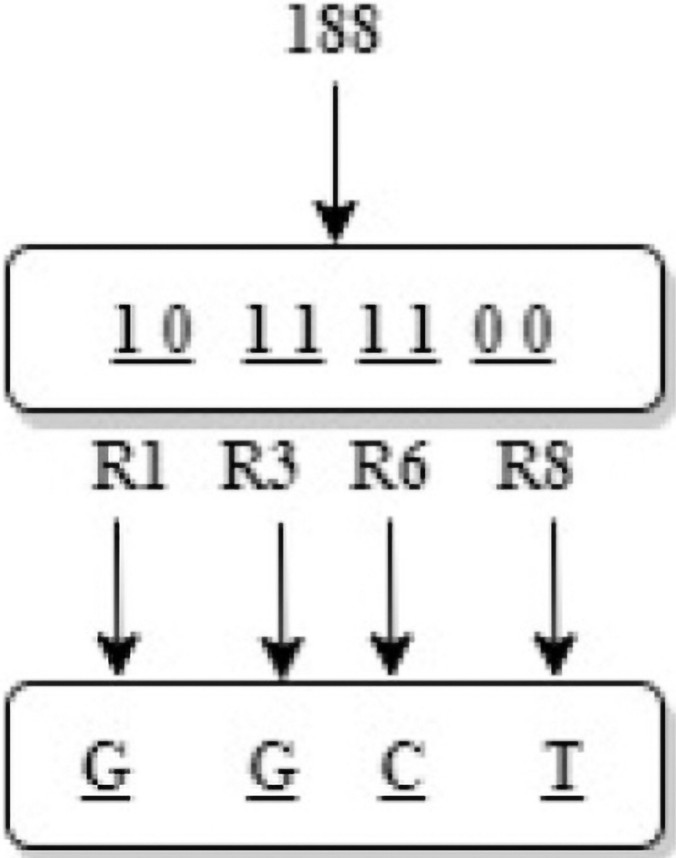

**Fig 5. Bit-level dynamic DNA coding process diagram.**

When each elements of matrix is calculated, a *DifMatrix* matrix with a size of 4*MN* is generated.

## 3.7 Dynamic DNA mutation

Dynamic DNA mutation is a process of dynamically changing each base through the selection of DNA mutation rules and the number of DNA mutations.

Step 1: Using the Logistic chaotic sequence, the key stream *Key* and the key feature value *Key-Feature*, a chaotic sequence *ChaoticSignal*2 with a length of 4*MN* is generated. Since there are 6 DNA mutation rules, the number of DNA mutations can be controlled within 1-3, so the DNA mutation rule $R1$ and the number of DNA mutations $N1$ are generated as shown in Eqs (22) and (23).

$$R1 = floor(6 * ChaoticSignal2) + 1 \qquad (22)$$

$$N1 = floor(3 * ChaoticSignal2) + 1 \qquad (23)$$

Among them, *floor*() is a round-down function.

Step 2: This algorithm stipulates that when $N1$ is an odd number, first perform mutations according to the DNA mutation rule $R1$, and then perform DNA mutations according to the complementation rule 1 base direct complement rule; when $N1$ is an even number, directly perform mutations according to the DNA mutation rule $R1$. When each base is mutated according to the corresponding number of DNA mutations and DNA mutation rules, a *MutMatrix* matrix with a size of 4*MN* is generated.

## 3.8 Encryption algorithm

The specific encryption algorithm steps are described as follows:

1. Input the original image $I(M, N)$ and the known key stream *KeyHex*, and use the hash algorithm to generate the 32-bit decimal final key stream *Key*. The specific process is as shown in 3.2.1.

2. Each bit of *Key* is XOR with each other to obtain a decimal key feature value *KeyFeature*, as shown in 3.2.2.

3. Use *Key*, *KeyFeature*, Logistic chaotic system, five-dimensional three-leaf chaotic system and the principle of DNA recombination to generate a chaotic matrix *KeyMatrix* with a size of 4*MN*. The generation process is shown in 3.3.

4. Perform bit-level dynamic DNA coding on every 2-bit binary of the original image according to the DNA coding rule $R$, and get the coded matrix *DecodedMatrix* with a size of 4*MN*, as shown in 3.4.

5. Generate a chaotic sequence through *Key* and *KeyFeature* and Logistic chaos. The chaotic sequence is in ascending order to generate an index sequence, and the position of each pixel value is scrambled to obtain the scrambled matrix *PerMatrix*, as shown in 3.5.

6. *PerMatrix* and the chaotic matrix *KeyMatrix* dynamically perform DNA operations according to the DNA operation rules *Operation* to obtain the *DifMatrix* matrix. The detailed process is shown in 3.6.

7. The DNA mutation rule $R1$ and the number of DNA mutations $N1$ are used to dynamically determine how a certain base undergoes DNA mutation operations. When $N1$ is an odd number, first perform DNA mutation according to $R1$, and then perform direct complement operation; When $N2$ is an even number, perform DNA mutation directly according to the $R1$ rule. Finally, when all bases have completed the mutation process, a 4*MN* size *MutMatrix* matrix is generated. The specific steps are shown in 3.7.

8. Use part of the key stream in *Key* and *KeyFeature* to generate new Logistic initial values and parameters, and iterate 4*MN* times to generate chaotic sequences and convert them into DNA decoding rules. The *MutMatrix* matrix uses DNA decoding rules to dynamically decode DNA to obtain the final encrypted image $C$, and the encryption is completed.

## 3.9 Decryption algorithm

The decryption algorithm is the inverse process of the encryption algorithm. Among them, it should be noted that the reverse operation of DNA encoding is DNA decoding; DNA operations also have many reversible operations, such as DNA left shift and DNA right shift are

reversible; DNA mutation decryption algorithm, odd-numbered complementary operations need to perform complementary operation 2, and then complementary operation 1; Even-numbered complementary operations can directly perform complementary operation 1; In addition, the decryption process also needs to pay attention to the number of complements of complementary operation 1. Other operations are similar to encryption and will not be described here.

## 4 Simulation results and security analysis

### 4.1 Simulation results

The key of this algorithm includes the initial key stream and the control parameters of the five-dimensional three-leaf chaotic system. The keys used in this algorithm are shown in Table 11.

In the Win10 system, this algorithm uses MATLAB2020a software to perform encryption and decryption simulation experiments on 3 different medical images of MRI, CT and X-ray. The three forms of medical images used in this paper have a total of 12 images. Each medical image is selected from different parts of the human body, and the original image, encrypted image and decrypted image are shown in the table. All images are from https://openmd.com/. The experimental results are shown in Figs 6–8.

According to the experimental results, it can be seen that the encrypted image to be decrypted image has become indistinguishable snowflake noise, and the encrypted image has no relationship with the decrypted image; through the decryption algorithm, there is no visual difference between the decrypted image and the original image. On the whole, it meets the encryption and decryption requirements.

The safety analysis in the following sections takes the Axillary image of MRI as an example. The algorithm is analyzed and tested in several sections including analysis of decryption algorithm restoration ability, key space analysis, information entropy, histogram, correlation analysis, plaintext sensitivity analysis, key sensitivity analysis, noise attack, cropping attack, and time performance.

### 4.2 Analysis of decryption algorithm restoration ability

A complete medical image encryption algorithm is divided into an encryption part and a decryption part. Therefore, the decryption algorithm also occupies an important position in the entire algorithm. The decryption algorithm should ensure that the encrypted image can be restored to the greatest extent after being decrypted by the decryption algorithm. The degree of restoration of the decryption algorithm can be seen visually and intuitively, and can also be reflected in the data. In Section 4.1, the decryption and restoration ability of this algorithm can be confirmed visually. This section uses the similarity between the decrypted image and the original image and PSNR to measure the restoration ability of the decryption algorithm at the data level [46]. The definition of PSNR is given below as shown in Eqs (24) and (25).

$$MSE = \frac{1}{mn} \sum_{i=0}^{m-1} \sum_{j=0}^{n-1} ||I(i,j) - K(i,j)||^2 \tag{24}$$

**Table 11. Key table of this algorithm.**

| Composition of the key | The key of encryption and decryption |
| --- | --- |
| Hash value | 527b9b3c77826d30a79e612114a8c 18df984c176f4e529f684748ad052241b17 |
| Control parameters of five-dimensional three-leaf chaotic system | a = -1;b = -3;c = -3;d = 1.8;e = -5 |

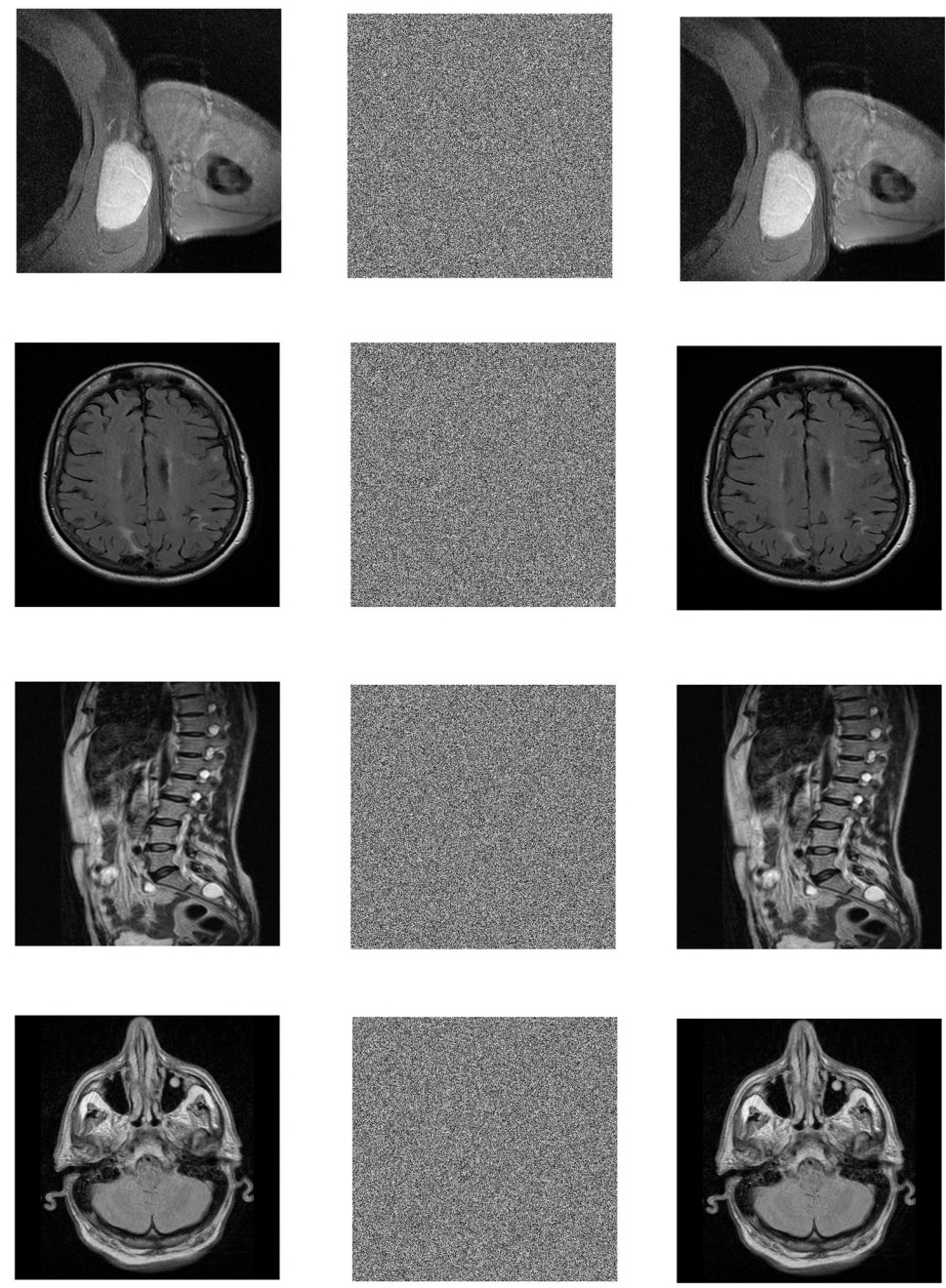

**Fig 6. Simulation results of MRI images.** (a) Axillaty encrypted image, (b) Axillaty encrypted image, (c) Axillaty decrypted image, (d) Brain original image, (e) Brain encrypted image, (f) Brain decrypted image, (g) Lumbar original image, (h) Lumbar encrypted image, (i) Lumbar decrypted image, (j) Patient brain original image, (k) Patient brain encrypted image, (l) Patient brain decrypted image.

$$PSNR = 10 * log_{10}(\frac{MAX_I^2}{MSE})\qquad(25)$$

Among them, *MSE* represents the mean square error of the current image *I* and the reference image *K*. *m* and *n* respectively represent the height and width of the image, and *I*(*i*, *j*) and

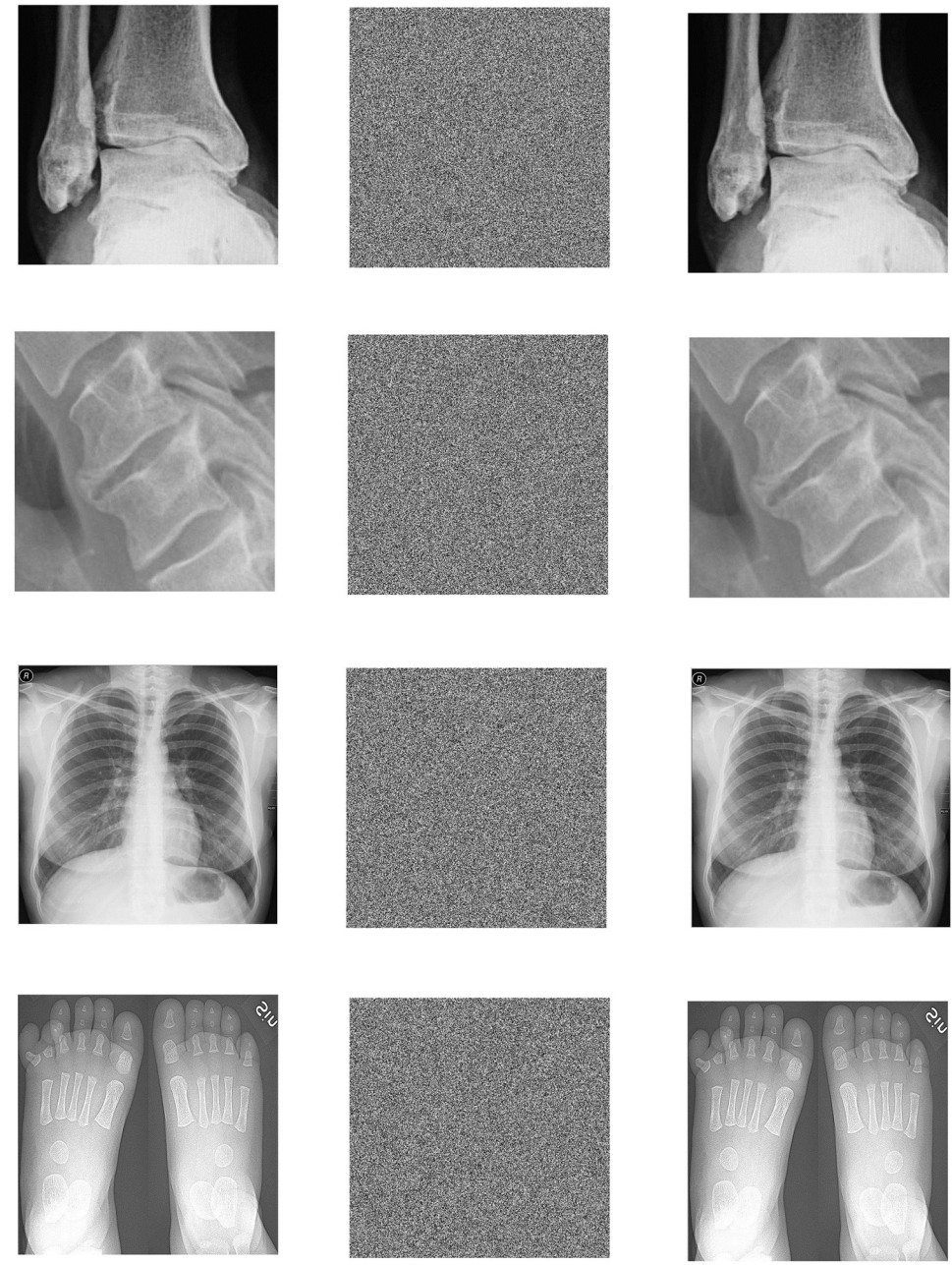

**Fig 7. Simulation results of X-ray images.** (a) Ankle original image, (b) Ankle encrypted image, (c) Ankle decrypted image, (d) Cervical original image, (e) Cervical encrypted image, (f) Cervical decrypted image, (g) Chest original image, (h) Chest encrypted image, (i) Chest decrypted image, (j) Feet original image, (k) Feet encrypted image, (l) Feet decrypted image.

$K(i, j)$ represent the pixel value at the corresponding coordinates. $MAX_I$ represents the maximum possible pixel value of the image. If each pixel is represented by 8-bit binary, then it is 255.

Select one image from the three types of medical images of MRI, CT and X-ray respectively, and calculate the similarity and PSNR value between the original image and the decrypted image, as shown in Table 12.

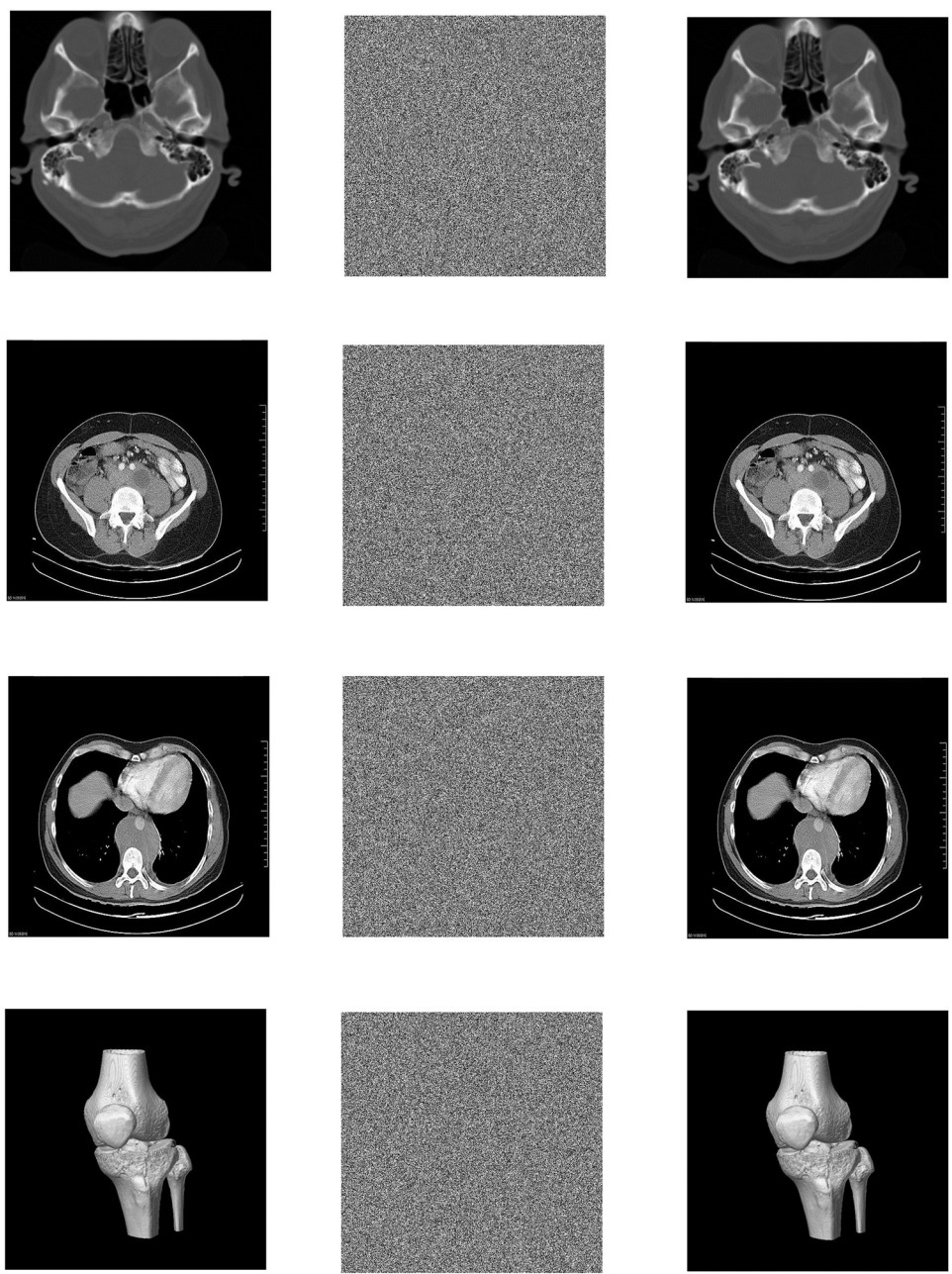

**Fig 8. Simulation results of CT images.** (a) Brain original image, (b) Brain encrypted image, (c) Brain decrypted image, (d) Old man testicular original image, (e) Old man testicular encrypted image, (f) Old man testicular decrypted image, (g) Testicular original image, (h) Testicular encrypted image, (i) Testicular decrypted image, (j) Tibial original image, (k) Tibial encrypted image, (l) Tibial decrypted image.

In general, the higher the similarity between the decrypted image and the original image, the greater the PSNR, indicating that the decryption algorithm has a stronger ability to restore. It can be seen from Table 12 that the similarities of the three selected medical images of different types are all 100%, and the PSNR values are all infinite. Therefore, the data shows that there is a very small difference between the decrypted image and the original image, which is

**Table 12. Comparison table of similarity and PSNR.**

| Type-Image name | MRI-Axillaty | X-ray-Ankle | CT-Brain |
|---|---|---|---|
| Similarity (%) | 100% | 100% | 100% |
| PSNR(dB) | $+\infty$ | $+\infty$ | $+\infty$ |

almost negligible. In summary, the decryption algorithm of this medical image encryption algorithm has a strong ability to restore.

### 4.3 Key space analysis

Since the algorithm key mainly includes the 128-bit initial key stream and the control parameters of the five-dimensional three-leaf chaotic system, the control parameters of the five-dimensional three-leaf chaotic system are $a$, $b$, $c$, $d$, and $e$. The precision of each is $10^{14}$, and the total key space is $(10^{14})^5 = 10^{70} \approx 22^{10}$. Therefore, the key space of the algorithm must be greater than $2^{128}$. At present, the key space [46–51] is greater than or equal to $2^{100}$ to effectively resist brute force attacks. Therefore, the key space of this algorithm is large enough to effectively resist brute force attacks.

### 4.4 Information entropy

The image information entropy [52] reflects the distribution of image gray value. Its expression is as shown in Eq (26).

$$H(m) = -\sum_{i=0}^{L} P(m_i) log_2 P(m_i) \tag{26}$$

Among them, $m_i$ represents the $i$-th gray value of the $L$-level gray level, and $P(m_i)$ represents the probability of the appearance of $m_i$.

Ideally, the information entropy should be close to 8 [53]. The higher the information entropy, the more uniform the gray value distribution. The information entropy obtained by this algorithm is shown in Table 13.

The level of information entropy is an important indicator to measure the effect of encryption. MI in the Table 13 is the abbreviation of medical image. It can be seen from Table 13 that the information entropy of the encrypted image obtained by this algorithm has been significantly improved. The information entropy of this algorithm is slightly higher than that of the reference, and overall it is close to 8. Therefore, this encryption algorithm is slightly better than the algorithms in other references in terms of information entropy.

### 4.5 Histogram

The histogram [55, 56] reflects the uniformity of the gray value distribution of the image. It can be seen from Fig 9. that the histogram of the original image is uneven, with high and low,

**Table 13. Information entropy comparison table.**

| Information entropy | proposed | Ref. [47](MI) | Ref. [48](MI) | Ref. [49](MI) | Ref. [1] | Ref. [33] | Ref. [54] |
|---|---|---|---|---|---|---|---|
| original image | 7.235847 | 7.235847 | 7.235847 | 7.235847 | 7.7548 | 7.445507 | 7.5929 |
| encrypted image | 7.999341 | 7.998284 | 7.9973 | 7.9992 | 7.999196 | 7.9966 | 7.9993 |

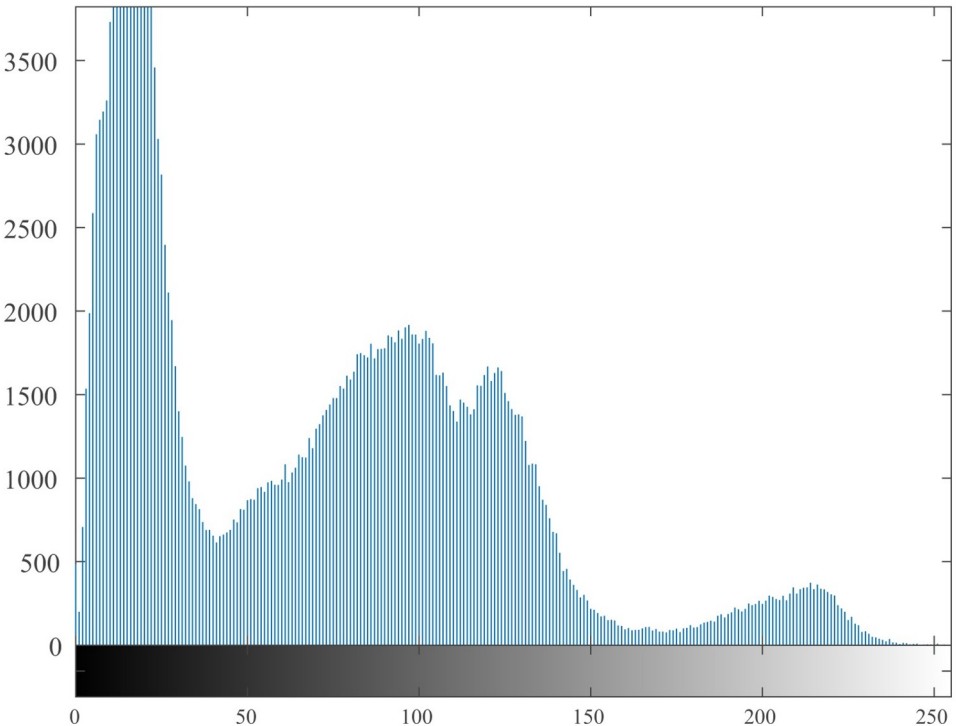

**Fig 9. Histogram of the original image.**

and it is easy to be cracked by statistical attacks. The histogram of the encrypted image is shown in Fig 10. It can be seen from the figure that its histogram is evenly distributed, which reduces the risk of being cracked by statistical attacks.

It can be seen from the histogram illustrations of the original image and the encrypted image that the histogram obtained by the encryption algorithm of this algorithm is more uniform, indicating that the algorithm can effectively resist statistical attacks.

## 4.6 Correlation coefficient and point image

Correlation coefficient [57, 58] is used to measure the correlation degree of adjacent pixels of an image. The definition of correlation coefficient is as shown in Eqs (27)–(30).

$$E(x) = \frac{1}{N} \times \sum_{i=1}^{N} x_i \tag{27}$$

$$D(x) = \frac{1}{N} \times \sum_{i=1}^{N} (x_i - E(x)^2) \tag{28}$$

$$cov(x) = \frac{1}{N} \times \sum_{i=1}^{N} (x_i - E(x))(y_i - E(y)) \tag{29}$$

$$r_{xy} = \frac{cov(x, y)}{\sqrt{D(x)}\sqrt{D(y)}} \tag{30}$$

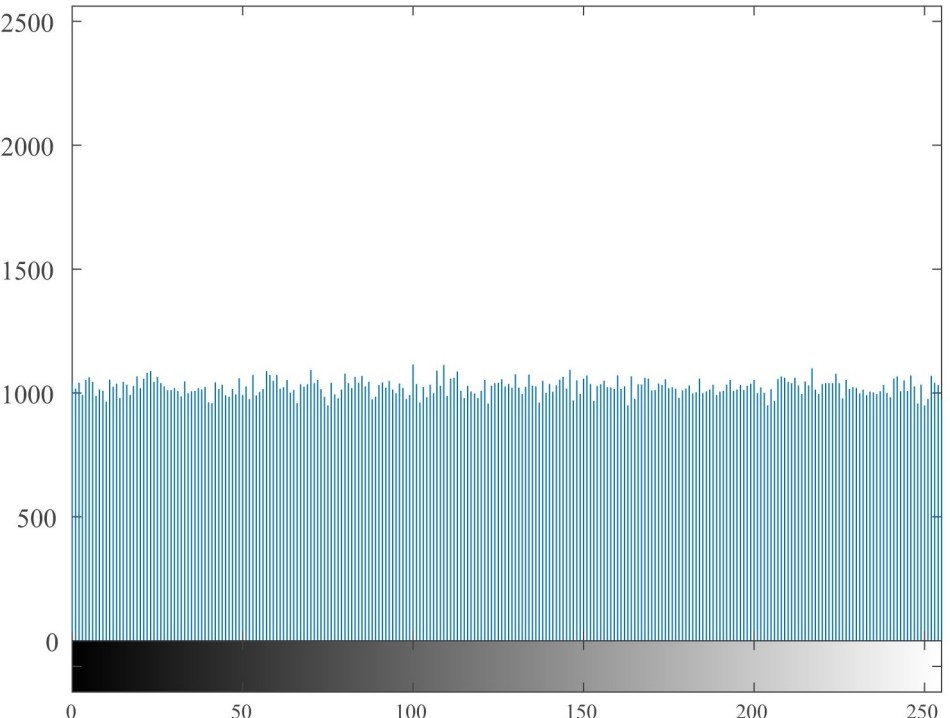

**Fig 10. Histogram of the encrypted image.**

Among them, $x$ and $y$ represent the gray values of two adjacent pixels in the image, and $N$ represents the logarithm of randomly selected pixels from the image.

The correlation coefficient between adjacent pixels in the original image is very high. After this encryption algorithm, the correlation coefficient is reduced. Ideally, the correlation coefficient should be close to 0 [59]. 1000 pairs of pixel values are randomly selected, and the correlation coefficient comparison are shown in Tables 14 and 15.

Compared with the correlation coefficient of the medical image in Table 14, the correlation coefficient of the encrypted image obtained by this algorithm is slightly lower than the [47, 48], and slightly higher than the [49]. Compared with the correlation coefficient of ordinary images in Table 15, the correlation coefficient of the encrypted image obtained by this algorithm is slightly lower than that of [1], and slightly higher than that of [33, 54]. It shows that the algorithm can effectively reduce the correlation between adjacent pixels. Whether compared with the reference literature of ordinary image encryption or the reference literature of medical image encryption, the ability to resist statistical attacks is slightly stronger.

The distribution of adjacent pixels is shown in Fig 11. Among them (a), (b), (c) are the correlation point image of the original image in 3 directions, (d), (e), (f) are the correlation point diagrams of the encrypted image in 3 directions.

**Table 14. Medical image correlation coefficient comparison table.**

|  | horizontal | vertical | diagonal | Ref.[47](MI) | Ref.[48](MI) | Ref.[49](MI) |
|---|---|---|---|---|---|---|
| original image | 0.9954 | 0.9973 | 0.9932 | 0.92497 | 0.9953 | 0.97087 |
| encrypted image | -0.0023 | 0.0014 | 0.0025 | 0.004267 | 0.00753 | 0.0022859 |

**Table 15. Ordinary image correlation coefficient comparison table.**

|  | horizontal | vertical | diagonal | Ref.[1] | Ref.[33] | Ref.[54] |
|---|---|---|---|---|---|---|
| original image | 0.9954 | 0.9973 | 0.9932 | 0.97331 | 0.9707 | 0.978 |
| encrypted image | -0.0023 | 0.0014 | 0.0025 | 0.0013702 | 0.0106 | 0.01397 |

It can be seen from Fig 11 that the correlation point map of the original image is distributed in a straight line, close to the straight line. After this encryption algorithm, the correlation point map of the encrypted image is evenly distributed, which effectively resists statistical attacks.

## 4.7 Plaintext sensitivity analysis

A good encryption algorithm should be sensitive to the plaintext [60]. If the plaintext is changed a little bit, the ciphertext will change greatly. Usually use NPCR, UACI [61] to analyze the difference between two images. It is defined as shown in Eqs (31)–(33).

$$NPCR = \frac{\sum_{i=0}^{M-1} \sum_{j=0}^{N-1} D(i,j)}{M \times N} \times 100\% \tag{31}$$

$$D(i,j) = \begin{cases} 0, E(i,j) = E'(i,j) \\ 1, E(i,j) \neq E'(i,j) \end{cases} \tag{32}$$

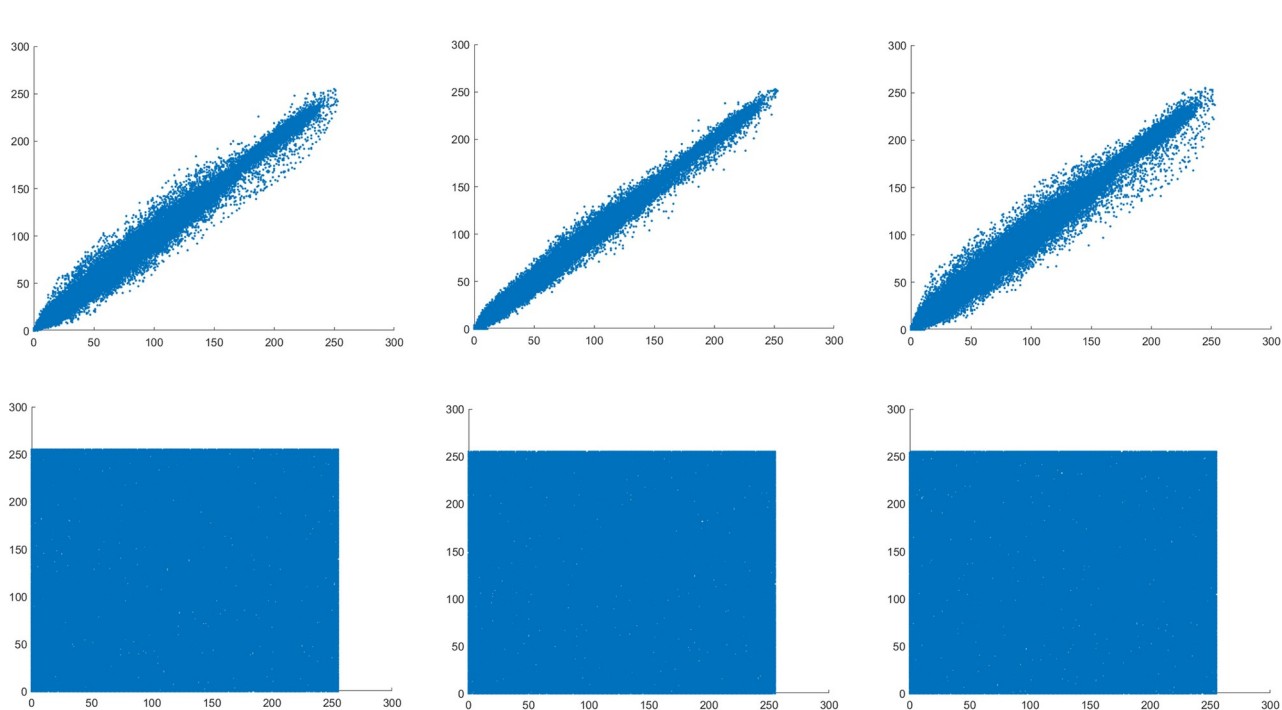

**Fig 11. Horizontal, vertical, diagonal correlation point diagrams of original image and encrypted image.** (a) original image horizontal, (b) original image vertical, (c) original image diagonal, (d) encrypted image horizontal, (e) encrypted image vertical, (f) encrypted image diagonal.

**Table 16. Comparison table of NPCR and UACI with randomly changing 1 pixel.**

| Pixel coordinates | NPCR | UACI |
|---|---|---|
| (15,10) | 99.6082% | 33.4828% |
| (149,105) | 99.6265% | 33.5387% |
| (506,478) | 99.6284% | 33.5387% |

$$UACI = \frac{\sum_{i=0}^{M-1} \sum_{j=0}^{N-1} \frac{|E(i,j) - E'(i,j)|}{255}}{M \times N} \times 100\% \tag{33}$$

Among them, $E(i, j)$ and $E'(i, j)$ respectively represent the pixel gray value of two ciphertext images at coordinates $(i, j)$; $M$ and $N$ represent the height and width of the image respectively. $D(i, j)$ is defined as follows: if $E(i, j) = E'(i, j)$, then $D(i, j) = 1$; otherwise, $D(i, j) = 0$.

$NPCR$ = 99.6094% and $UACI$ = 33.4635% are the expected values [62]. The larger the value of $NPCR$ and $UACI$, the greater the difference between ciphertexts.

The first group randomly selects one pixel point and adds 1 to the pixel value of the point, and conducts 3 experiments randomly. The comparison of NPCR and UACI of the encrypted image is shown in Table 16.

In the second group, two pixel points were randomly selected, the first point was added by 1 based on the original pixel value, and the second point was subtracted by 1 based on the original pixel value, and 3 experiments were performed randomly. The comparison of NPCR and UACI of the encrypted image is shown in Table 17.

From the comparison table of NPCR and UACI with randomly changing 1 pixel value and randomly changing 2 pixel values, it can be seen that the values of NPCI and UACI fluctuate slightly above and below the ideal value, and most experimental comparisons are higher than the ideal value. Therefore, it can be explained that there is a big difference between the encrypted image generated by changing the pixel value of the original image and the encrypted image generated without changing the original image. The algorithm has better plaintext sensitivity.

## 4.8 Key sensitivity analysis

Key sensitivity [55–57] means that when the key is slightly changed, it will cause a big change, and the correct decrypted image cannot be obtained through the decryption algorithm.

The original 32-bit final key stream is {224, 107, 117, 68......254, 159, 34}. This test changes the last bit of the key stream from 34 to 35 on the basis of the original 32-bit key stream. The changed 32-bit final key stream is {224, 107, 117, 68......254, 159, 34}.

**Table 17. Comparison table of NPCR and UACI with randomly changing 2 pixel.**

| Pixel coordinates | NPCR | UACI |
|---|---|---|
| (15,10) (46,224) | 99.6155% | 33.4095% |
| (149,105) (323,89) | 99.6101% | 33.4995% |
| (506,478) (173,285) | 99.6098% | 33.4365% |

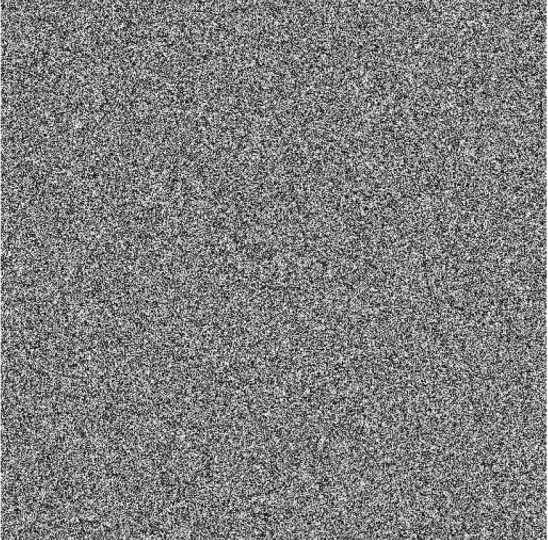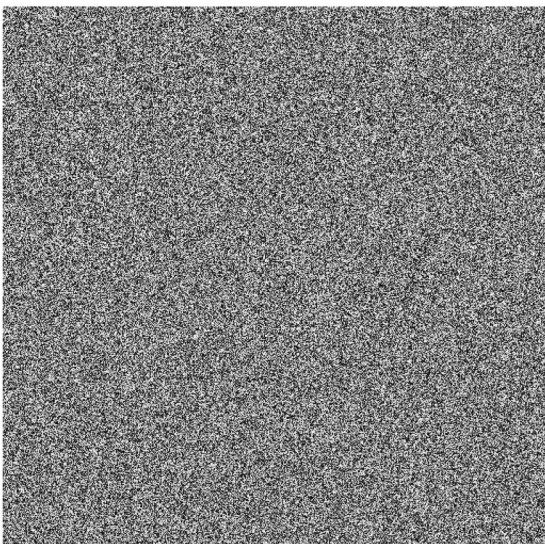

**Fig 12. Encrypted and decrypted image after changing the key.** (a) encrypted image, (b) decrypted image.

This test uses the changed key stream to decrypt the original encrypted image to obtain the decrypted image. The comparison between the original encrypted image and the decrypted image obtained by changing the key is shown in Fig 12.

It can be seen from Fig 12 that when the last bit of the original 32-bit final key stream is changed randomly, the visually unrecognizable snowflake noise image can be obtained. It can be concluded that when the key changes slightly, the correct decrypted image cannot be obtained, indicating that the algorithm has strong key sensitivity.

### 4.9 Noise attack analysis

In the process of widespread transmission of images, it is inevitable that they will be affected and destroyed by noise. A good encryption algorithm must ensure that the image can still get a visually identifiable decrypted image after being attacked by noise [63, 64].

The encrypted image is increased by 0.005, 0.05 and 0.1 salt and pepper noise attacks. After the decryption algorithm, the effect of the decrypted image is shown in Fig 13.

It can be seen from Fig 13 that when different degrees of salt and pepper noise attacks are added to the encrypted image, although the clarity of the decrypted image will be reduced, it will not affect the normal image recognition. This proves that this algorithm can effectively resist noise attacks of varying degrees.

### 4.10 Cropping attack analysis

As the image is transmitted on the public channel, it will inevitably suffer data damage, leading to the loss of image information. A good encryption algorithm must ensure that the damaged image information can still be visually recognized through the decryption process [65, 66].

The encrypted image is cropped by 1/16, 1/4, 1/2 from the upper left corner, and the cropped part is replaced with 0. The cropped encrypted image is shown in Fig 14. The encrypted image is decrypted through the decryption algorithm to obtain the decrypted image. The effect is as follows shown in Fig 15.

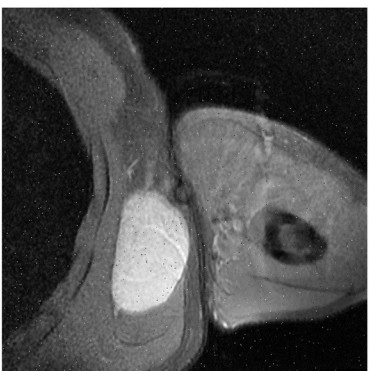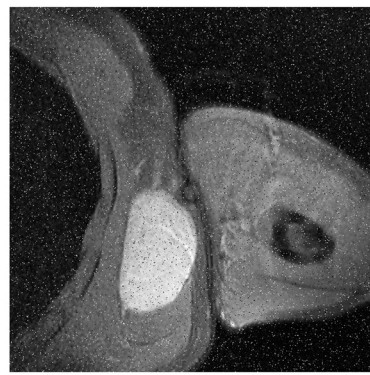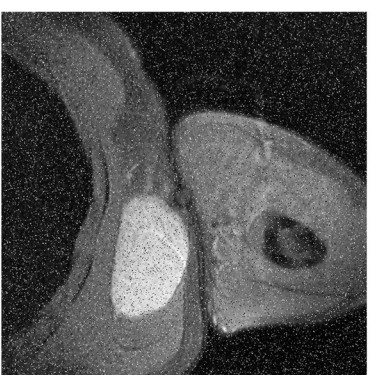

**Fig 13. Decrypted images of different levels of salt and pepper noise attacks.** (a) 0.005 salt and pepper noise, (b) 0.05 salt and pepper noise, (c) 0.1 salt and pepper noise.

After the encrypted image is cropped with different sizes, the cropped encrypted image is decrypted by this algorithm. It can be seen from Fig 15 that although the clarity of the obtained decrypted image is affected, the overall recognition of the decrypted image will not be achieved. This proves that the algorithm can resist cropping attacks to a certain extent.

## 4.11 Time performance analysis

A good algorithm must ensure that the encrypted information can be obtained in a short time, and the encrypted image information can be decrypted within a valid time [4, 47, 67]. The comparison of the encryption and decryption time of this algorithm with other algorithms is shown in Table 18.

In the Table 18, use MI instead of medical image. It can be seen from Table 18 that the encryption and decryption time of this algorithm is slightly longer than that of [48], which is slightly shorter than the other 3 references. The encryption and decryption time can be controlled at about 1 second. It shows that this algorithm has higher time efficiency and stronger real-time performance than most references.

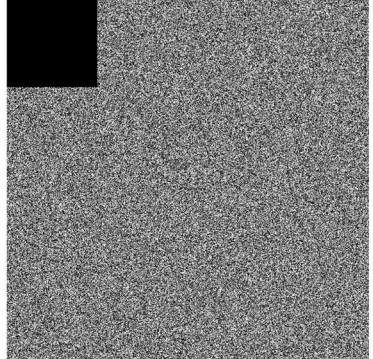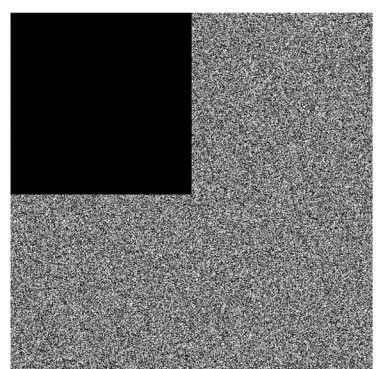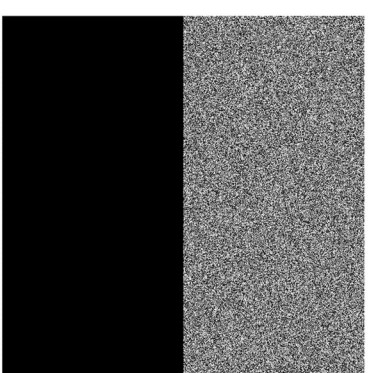

**Fig 14. Encrypted image after cropping.** (a) 1/16 cropped image, (b) 1/4 cropped image, (c) 1/2 cropped image.

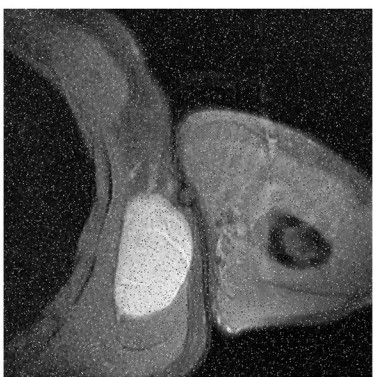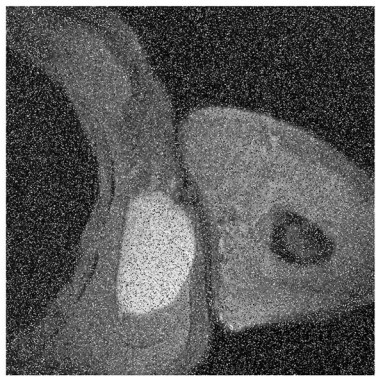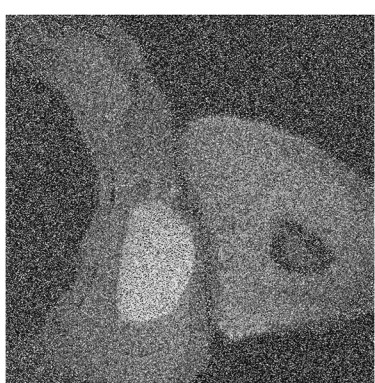

**Fig 15. Decrypted image after cropping attack.** (a) 1/16 decrypted image, (b) 1/4 decrypted image, (c) 1/2 decrypted image.

### 4.12 Analysis of the effect of ordinary image encryption

This medical image encryption algorithm [49] can not only perform safe and efficient encryption and decryption of medical images with high resolution and large redundancy, but also applies to ordinary image encryption. An image from https://unsplash.com is selected for encryption and decryption, and the original image, encrypted image, decrypted image, and histograms corresponding to the three sets of images are shown in Fig 16.

It can be seen from Fig 16 that there is no visual difference between the ordinary original image and the decrypted image, the encrypted image becomes a snowflake noise image, and it is no longer visible that the original image and the encrypted image have any relevant information. The histogram of the original image and the histogram of the decrypted image are regularly distributed according to the information characteristics, while the gray value of the histogram of the encrypted image is evenly distributed, showing a linear distribution. It shows that after the ordinary image is encrypted, the ability of the encrypted image to resist statistical attacks is enhanced. In summary, this medical image encryption algorithm is suitable for ordinary image encryption, and the encryption effect is good.

## 5 Conclusion

This paper proposes a medical image encryption algorithm based on a new five-dimensional three-leaf chaotic system and genetic operation. The introduction of the hash algorithm increases the key space and the correlation between the original image and the key; Based on the new five-dimensional three-leaf chaotic system and the principle of DNA recombination, it improves the randomness of the chaotic matrix while also increasing the randomness of the encryption algorithm; The addition of dynamic DNA mutation operation not only increases the biological significance, but also each base will be randomly changed; Using bit-level dynamic DNA coding, each 2 bits of binary code is coded with different coding rules, and the diffusion effect is better; 7 kinds of DNA operation is expanded on the basis of basic DNA addition and subtraction, making the operation of each pixel value more random and

**Table 18. Comparison table of encryption and decryption time.**

|  | proposed | Ref. [48](MI) | Ref. [1] | Ref. [54] | Ref. [68] |
|---|---|---|---|---|---|
| Time(s) | 0.947331 | 0.94 | 1.363 | 1.0921 | 5.257 |

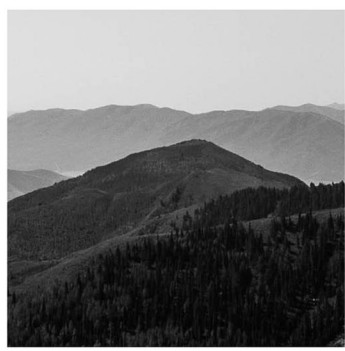 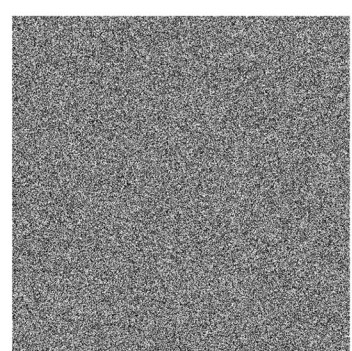 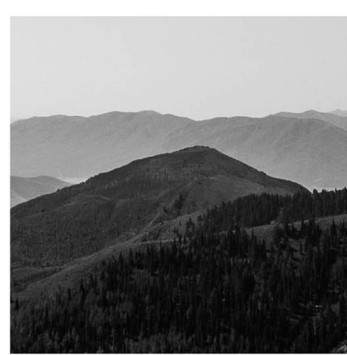

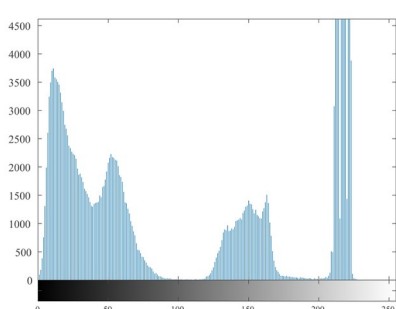 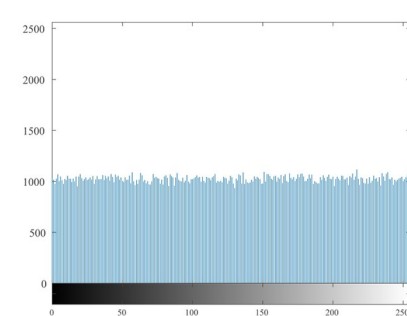 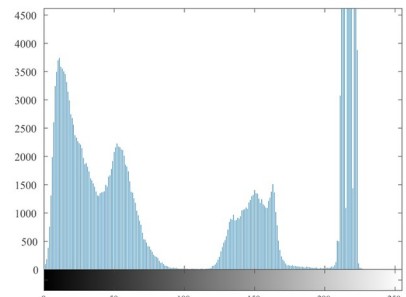

**Fig 16. Simulation experiment results of ordinary image.** (a) original image, (b) encrypted image, (c) decrypted image, (d) histogram of original image, (e) histogram of encrypted image, (f) histogram of decrypted image.

changeable. After security analysis and time efficiency analysis, it is shown that this medical image encryption algorithm is more secure and reliable than the latest medical image encryption algorithm, and has stronger real-time performance. In addition, the encryption and decryption experiments of ordinary images using this algorithm also show good encryption results. Therefore, this medical image encryption algorithm is suitable for both medical image encryption and ordinary image encryption.

## Author Contributions

**Conceptualization:** Zhongyue Liang.

**Formal analysis:** Zhongyue Liang, Qiuxia Qin, Yi Xu.

**Funding acquisition:** Changjun Zhou, Yi Xu, Wenshu Zhou.

**Methodology:** Zhongyue Liang, Qiuxia Qin, Changjun Zhou.

**Visualization:** Qiuxia Qin, Yi Xu, Wenshu Zhou.

**Writing – original draft:** Zhongyue Liang.

**Writing – review & editing:** Zhongyue Liang, Changjun Zhou, Ning Wang, Yi Xu.

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
