## [Decision Letter · Decision Letter 0]

27 Aug 2021

PONE-D-21-19375

Medical image encryption algorithm based on a new five-dimensional three-leaf chaotic system and genetic operation

PLOS ONE

Dear Dr. Xu,

Thank you for submitting your manuscript to PLOS ONE. After careful consideration, we feel that it has merit but does not fully meet PLOS ONE’s publication criteria as it currently stands. Therefore, we invite you to submit a revised version of the manuscript that addresses the points raised during the review process.

The paper presents a new higher order chaotic system with DNA recombination. It is well written and reads interesting. The author should clearly represent, perhaps at the section heading, their contribution, which is the proposed/new higher order chaotic system with clearer justification of the choice of using DNA recombination. The paper should be carefully revised to avoid any grammatical errors and typos.

The figures and tables in Sec. 4 are to be explained with more details and a summary of the analysis and comparisons is required to highlight the merit of the proposed system in support of the conclusions.

The reviewers' concerns should also be adequately addressed.

We look forward to receiving your revised manuscript.

Kind regards,

Hussain Md Abu Nyeem, Ph.D.

Academic Editor

PLOS ONE

Journal Requirements:

"This work was supported in part by the National Youth Science Foundation of China 490

under the Grant numbers 62002046, 61802040, and in part by the National Natural 491

Science Foundation of China (61672121)."

"National Youth Science Foundation of China(No.62002046, No.61802040).

National Natural Science Foundation of China

(No.61672121)."

Reviewers' comments:

Reviewer's Responses to Questions

**Comments to the Author**

1. Is the manuscript technically sound, and do the data support the conclusions?

Reviewer #1: Partly

Reviewer #2: Yes

Reviewer #3: Yes

2. Has the statistical analysis been performed appropriately and rigorously? 

Reviewer #1: N/A

Reviewer #2: Yes

Reviewer #3: Yes

3. Have the authors made all data underlying the findings in their manuscript fully available?

Reviewer #1: Yes

Reviewer #2: Yes

Reviewer #3: Yes

4. Is the manuscript presented in an intelligible fashion and written in standard English?

Reviewer #1: Yes

Reviewer #2: Yes

Reviewer #3: Yes

5. Review Comments to the Author

Reviewer #1: This paper proposes a medical image encryption algorithm based on a new five-dimensional three-leaf chaotic system and genetic operation. Simulation experiments show that the algorithm has good encryption effect, high time efficiency, and can effectively resist various attacks.

Some major revisions should be addressed for improving the quality of the paper.

1. There are some spelling and syntax errors, which need correction (e.g. “an medical image encryption algorithm” -> “a medical image encryption algorithm”).

2. After Eq. (1) and Eq. (2), “where” is necessary.

3. Fig. 2 is not a bifurcation diagram but several phase portraits. Besides, just two phase portraits will be enough.

4. In Line 170 and Eq. (10), “x(1), y(1), z(1), w(1), z(1)” -> “x(1), y(1), z(1), w(1), v(1)”.

5. Recently, the ISI-encoded chaotic sequences were applied to the application of image encryption in [Bao H, Hua ZY, Liu WB, et al. Discrete memristive neuron model and its interspike interval-encoded application in image encryption. Sci China Tech Sci, 2021, 64, 10.1007/s11431-021-1845-x]. Can these chaotic sequences be used for the work of this paper?

Reviewer #2: In this paper，chaotic system and DNA operations are applied to medical image encryption. The authors put forward some innovative ideas and methods, and the study has important reference significance for the research in related fields. However, there are some shortcomings that need to be modified by the author.

1. Figure 1 lacks the coordinate value and should be marked with a symbol representing the coordinate value.

2. Figure 2 is not a bifurcation diagram but a phase diagram of the new five-dimensional three-leaf chaotic

system.

3. There are some language expression problems that need to be improved, such as:

1) "Firstly, the original key stream and the original image call the hash algorithm to generate the final key stream", Improper use of the subject of the predicate verb call.

2) "the encrypted image is I" in row 139 page 6, it should be "the original image to be encrypted image is I".

3) Change "matrix point" to "elements of matrix" is better.

4) In row 310 of page 11, "the decrypted image ..." should be " the encrypted image...".

5) "The image obtained after encryption by this algorithm is shown in Fig7." should be "the histogram of the encrypted image is shown in Fig7.","Fig 7. Histogram of the encryption image." should be "Fig 7. Histogram of the encrypted image." ,"encryption image"->"encrypted image", "decryption image"->"decrypted image".

6) "Fig8."->"Fig. 8", and so on.

7) The first cell of column 1 of Table 17 and Table 18 should be filled with a name word.

Reviewer #3: The manuscript is well written but fails in justifications.

Please include the problem statement in the abstract.

By problem I mean Medical image encryption and it differences from image encryption.

Please include the reason that this approach is suited for Medical image encryption and maybe not for other kinds of encryption.

And again the approaches used such as five-dimensional three-leaf chaotic system. Why do you think that other chaotic maps can not suit this problem? The same for DNA operatons.

6. PLOS authors have the option to publish the peer review history of their article (what does this mean?). If published, this will include your full peer review and any attached files.

Reviewer #1: No

Reviewer #2: **Yes: **Congxu Zhu

Reviewer #3: No

---

## [Author Response · Author response to Decision Letter 0]

24 Sep 2021

Response to Reviewers

First of all, the authors would like to thank the editor and the reviewers for their careful reading of this paper and helpful comments, which have been very useful for improving the quality and presentation of this paper. 

According to the editor and reviewers’ comments, we have revised our manuscript. You will find that we have corrected the manuscript taking in the editor and reviewers’ suggestions. We apologize to the editor and the reviewers for all the troubles we have caused with this manuscript. The explanation of the modifications as well as corrections in this revision can be arranged as follows: 

Editor:

Response to the editor’s comments:

1)The author should clearly represent, perhaps at the section heading, their contribution, which is the proposed/new higher order chaotic system with clearer justification of the choice of using DNA recombination.

Response: The author agrees with the editor's suggestion. This algorithm does not clearly explain the reason for choosing a five-dimensional three-leaf chaotic system and DNA recombination operation. Therefore, the author first added Sec. 2.2.1 and Sec. 2.2.2 in Sec. 2.2 to analyze the dynamic characteristics of the five-dimensional three-leaf chaotic system and test the randomness of the chaotic sequence. In the analysis process, the five-dimensional three-leaf chaotic system is compared with other high-dimensional chaotic systems, and the reason for choosing this system is explained. In addition, the author gave a more detailed description of the DNA recombination paragraph in the introduction, and added a paragraph in Sec. 2.5 to explain the reasons for choosing DNA recombination operations. In conclusion, the suggestions given by the editor have been carefully revised.

2)The paper should be carefully revised to avoid any grammatical errors and typos.

Response: The author agrees with the editor's suggestion, and there are some spelling and grammatical errors in this article. Therefore, the author has corrected the spelling and grammatical errors in this article. The specific content of the amendment can be seen in the content of the response to the reviewer's comments.

3)The figures and tables in Sec. 4 are to be explained with more details and a summary of the analysis and comparisons is required to highlight the merit of the proposed system in support of the conclusions.

Response: The author agrees with the editor's opinion. The figures and tables in the Sec. 4 does not have a more detailed explanation and analysis to highlight the advantages of the algorithm. Therefore, an appropriate amount of supplementary explanations have been made in each subsection of the Sec. 4, and the specific revisions can be seen in the comments of the reviewers.

Reviewers: 

Reviewer #1:

Response to the reviewer #1’s comments:

1)There are some spelling and syntax errors, which need correction (e.g. “an medical image encryption algorithm” -> “a medical image encryption algorithm”).

Response: The author agrees with the comments given by reviewer 1. The “an medical image encryption algorithm” in line 4 of the abstract section on page 1, line 78 on page 3, and line 579 on page 21 have been changed to “ a medical image encryption algorithm”.

2)After Eq. (1) and Eq. (2), “where” is necessary.

Response: The author agrees with reviewer 1’s opinion that there is indeed no “where” in Eq. (1) and Eq. (2). Therefore, “where” is added to line 89 on page 3, and “where” is added to line 101 on page 4. And after reading this article carefully, add a sentence leading to the formula before each formula to make the formula reference more standardized.

3)Fig. 2 is not a bifurcation diagram but several phase portraits. Besides, just two phase portraits will be enough.

Response: The author agrees with reviewer 1 that it is a phase diagram instead of a bifurcation diagram, and it is unnecessary to use a lot of phase diagrams. Therefore, the author revises Sec. 2.2. First, change the bifurcation diagram in line 112 of page 4 to a phase diagram, and the title of Fig. 2 also becomes a phase diagram. Only the two phase diagrams x-y-z and z-ω-v are retained, and the contents of Fig. 2 have been replaced in the Figure folder.

4)In Line 170 and Eq. (10), “x(1), y(1), z(1), w(1), z(1)”-> “x(1), y(1), z(1), w(1), v(1)”.

Response: The author agrees with the opinion of reviewer 1, in the original manuscript line 170 “x(1), y(1), z(1), w(1), z(1)” and Eq. (10) “x(1), y(1), z(1), w(1), z(1)” does have a problem. Therefore, the author changed “x(1), y(1), z(1), w(1), z(1)” to “x(1), y(1), z(1), w(1), v(1)” in line 240 of page 9 of the revised manuscript. The last appearance of “z(1)” in Eq. (10) is changed to “v(1)”. 

5)Recently, the ISI-encoded chaotic sequences were applied to the application of image encryption in [Bao H, Hua ZY, Liu WB, et al. Discrete memristive neuron model and its interspike interval-encoded application in image encryption. Sci China Tech Sci, 2021, 64, 10.1007/s11431-021-1845-x]. Can these chaotic sequences be used for the work of this paper?

Response: First of all, thank you very much for recommending articles to us to enrich our articles. We have read this article carefully, which provides a new perspective for image encryption. First, a discrete mHR model is proposed, and then an ISI encoding algorithm is proposed. Using this algorithm, a chaotic sequence that can be applied to image encryption is generated. Subsequent writing of other articles will consider applying the content of this article, and this article has been introduced into this article. Again, thank you for your recommendation.

Reviewer #2:

Response to the reviewer #2’s comments:

1)Figure 1 lacks the coordinate value and should be marked with a symbol representing the coordinate value.

Response: The author agrees with reviewer 2’s opinion that Fig. 1 lacks coordinate value symbols to mark coordinate values. The author replaces Fig. 1 with coordinate value symbols for the original Fig. 1 in the folder Figure.

2)Figure 2 is not a bifurcation diagram but a phase diagram of the new five-dimensional three-leaf chaotic system.

Response: The author agrees with reviewer 2’s opinion that it is indeed a phase diagram rather than a bifurcation diagram. Therefore, the author revises Sec. 2.2. First, the author changed the bifurcation diagram in line 112 on page 4 to a phase diagram, and the title of Fig. 2 also became a phase diagram, and only retained the x-y-z and z-w-v phase diagrams to remove other redundant phase diagrams.

3)There are some language expression problems that need to be improved, such as:

a)“Firstly, the original key stream and the original image call the hash algorithm to generate the final key stream”, Improper use of the subject of the predicate verb call.

Response: The author agrees with reviewer 2’s opinion that the subject of the predicate verb is used improperly. The author revised this sentence again, and the revised sentence can be seen from lines 195 to 197 on page 8 of the revised manuscript.

b)“the encrypted image is I” in row 139 page 6, it should be “the original image to be encrypted image is I”.

Response: The author agrees with reviewer 2. Therefore, the author has made corresponding revisions in lines 207-208 on page 8 according to the revision tips of reviewer 2.

c)Change “matrix point” to “elements of matrix” is better.

Response: The author agrees with reviewer 2. Therefore, the author made amendments in accordance with the prompts given by the reviewer. There are 3 amendments in total, which are line 275 on page 10, line 313 on page 11, and line 315 on page 11.

d)In row 310 of page 11, “the decrypted image ...” should be “ the encrypted image...”.

Response: The author agrees with reviewer 2. Therefore, the author made corresponding amendments in line 386-387 on page 13 in accordance with the reviewer's amendment prompts.

e)“The image obtained after encryption by this algorithm is shown in Fig7.” should be “the histogram of the encrypted image is shown in Fig7.”,“Fig 7. Histogram of the encryption image.” should be “Fig. 7 Histogram of the encrypted image.” ,“encryption image”->“encrypted image”, “decryption image”->“decrypted image”.

Response: The author agrees with reviewer 2. First of all, the author is in line 448-449 on page 17, and the title of Fig. 7 has been revised according to the reviewer’s revision prompts. Secondly, the author changed all “encryption image” to “encrypted image” and all “decryption image” to “decrypted image”, and the revised manuscript marked all changes.

f)“Fig8.”->“Fig. 8”, and so on.

Response: The author checked the latex template downloaded from the official website based on the reviewer's comments, but the format generated by the template used is fixed. Neither the title nor the main text gives the format suggested by the reviewer. So we still use the format in the official website template. The format in the official website template is shown in the figure below.

g)The first cell of column 1 of Table 17 and Table 18 should be filled with a name word. 

Response: The author agrees with reviewer 2’s opinion that the first cell in column 1 of Table 17 and Table 18 has no name word. Table 17 and Table 18 suggested by the reviewer are now Table 19 and Table 20. Therefore, the author has added name words to the first cell of column 1 of Table 19 on page 19 and the first cell of column 1 of Table 20 on page 19. 

Reviewer #3:

Response to the reviewer #3’s comments:

1)Please include the problem statement in the abstract. By problem I mean Medical image encryption and it differences from image encryption. Please include the reason that this approach is suited for Medical image encryption and maybe not for other kinds of encryption.

Response: The author agrees with reviewer 3. First of all, medical image encryption is different from ordinary image encryption. Medical image encryption has higher requirements for the security, reliability and time efficiency of the algorithm. For security and reliability, on the one hand, various security tests such as key space, information entropy, and histogram are carried out in Sec. 4. On the other hand, it should reflect the accurate restoration ability of the decryption algorithm. Sec. 4.2 is added to illustrate this problem. For time efficiency analysis, the article has compared the latest medical image encryption algorithm and common image encryption algorithm in Sec. 4.11, and the result proves that this algorithm has strong real-time performance. Secondly, medical image encryption is different from ordinary image encryption, but the algorithm that can carry out medical image encryption can also carry out ordinary image encryption to a certain extent. Sec. 4.12 shows the encryption effect of this algorithm on ordinary image encryption. Finally, in the abstract and conclusion of this article, the above new content is supplemented to make the context correspond.

2)And again the approaches used such as five-dimensional three-leaf chaotic system. Why do you think that other chaotic maps can not suit this problem?

Response: The author agrees with reviewer 3. The author clarified the reasons for choosing the chaotic system from two aspects: the dynamic characteristics analysis of the five-dimensional three-leaf chaotic system and the random analysis of the chaotic sequence. First of all, the author adds Sec. 2.2.1, combined with the article that first proposed the chaotic system, and carried out dynamic analysis of the phase diagram, bifurcation diagram, balance points and other dynamics of the five-dimensional three-leaf chaotic system. Secondly, a new Sec. 2.2.2 uses the NIST test to test the randomness of the chaotic sequence and compare it with the NIST P value of the high-dimensional chaotic system. According to the experimental results, it can be known that the randomness of both the dynamic characteristics and the chaotic sequence is higher than that of most high-dimensional chaotic systems, and the author who proposed the system also applied the chaos to the image encryption algorithm to further illustrate the practicability of the chaotic system .

3)The same for DNA operatons.

Response: The author agrees with reviewer 3. First of all, the author has made some supplements in the introduction to the paragraphs that introduce the principles of DNA mutation and DNA recombination, indicating that DNA mutation and recombination operations enrich the encryption technology. In addition, the author added a paragraph in Sec. 2.4 to explain the reasons for using DNA mutation operations, and a paragraph in Sec. 2.5 to illustrate the reasons for using DNA recombination operations. In conclusion, the two DNA operations improve the scrambling and diffusion effects at the bit level, and the robustness and security of the algorithm are better guaranteed.

---

## [Decision Letter · Decision Letter 1]

2 Nov 2021

Medical image encryption algorithm based on a new five-dimensional three-leaf chaotic system and genetic operation

PONE-D-21-19375R1

Dear Dr. Xu,

We’re pleased to inform you that your manuscript has been judged scientifically suitable for publication and will be formally accepted for publication once it meets all outstanding technical requirements.

Kind regards,

Hussain Md Abu Nyeem, Ph.D.

Academic Editor

PLOS ONE

Reviewers' comments:

Reviewer's Responses to Questions

**Comments to the Author**

1. If the authors have adequately addressed your comments raised in a previous round of review and you feel that this manuscript is now acceptable for publication, you may indicate that here to bypass the “Comments to the Author” section, enter your conflict of interest statement in the “Confidential to Editor” section, and submit your "Accept" recommendation.

Reviewer #1: All comments have been addressed

Reviewer #2: All comments have been addressed

Reviewer #3: All comments have been addressed

Reviewer #4: All comments have been addressed

Reviewer #5: All comments have been addressed

2. Is the manuscript technically sound, and do the data support the conclusions?

Reviewer #1: Yes

Reviewer #2: Yes

Reviewer #3: Yes

Reviewer #4: No

Reviewer #5: Yes

3. Has the statistical analysis been performed appropriately and rigorously? 

Reviewer #1: Yes

Reviewer #2: Yes

Reviewer #3: Yes

Reviewer #4: Yes

Reviewer #5: Yes

4. Have the authors made all data underlying the findings in their manuscript fully available?

Reviewer #1: Yes

Reviewer #2: Yes

Reviewer #3: Yes

Reviewer #4: Yes

Reviewer #5: Yes

5. Is the manuscript presented in an intelligible fashion and written in standard English?

Reviewer #1: Yes

Reviewer #2: Yes

Reviewer #3: Yes

Reviewer #4: Yes

Reviewer #5: Yes

6. Review Comments to the Author

Reviewer #1: All my commends have been addressed in the revised manuscript. This paper can be accepted in the current version.

Reviewer #2: (No Response)

Reviewer #3: The authors have addressed all the comments and the responses were satisfying, therefore I suggest the article for being accepted.

Reviewer #4: (No Response)

Reviewer #5: the authors have satisfactorily addressed all the suggested comments.

to improve further, Give more justification - More comparison

7. PLOS authors have the option to publish the peer review history of their article (what does this mean?). If published, this will include your full peer review and any attached files.

Reviewer #1: No

Reviewer #2: No

Reviewer #3: **Yes: **Mohammad H Moattar

Reviewer #4: No

Reviewer #5: No

---

## [Editor Report · Acceptance letter]

16 Nov 2021

PONE-D-21-19375R1 

Medical image encryption algorithm based on a new five-dimensional three-leaf chaotic system and genetic operation 

Dear Dr. Xu:

I'm pleased to inform you that your manuscript has been deemed suitable for publication in PLOS ONE. Congratulations! Your manuscript is now with our production department. 

Kind regards, 

on behalf of

Dr. Hussain Md Abu Nyeem 

Academic Editor

PLOS ONE